# Common and distinct transcriptional signatures of mammalian embryonic lethality

John E. Collins[1,12], Richard J. White [1,2,12], Nicole Staudt[1], Ian M. Sealy [1,2], Ian Packham[1], Neha Wali[1], Catherine Tudor [1], Cecilia Mazzeo[1], Angela Green[1], Emma Siragher[1], Edward Ryder[1], Jacqueline K. White[1,9], Irene Papatheodoru [3], Amy Tang[3], Anja Füllgrabe [3], Konstantinos Billis [3], Stefan H. Geyer[4], Wolfgang J. Weninger[4], Antonella Galli[1], Myriam Hemberger [5,6,10], Derek L. Stemple [1,11], Elizabeth Robertson [7], James C. Smith [8], Timothy Mohun[8], David J. Adams[1] & Elisabeth M. Busch-Nentwich [1,2]

The Deciphering the Mechanisms of Developmental Disorders programme has analysed the morphological and molecular phenotypes of embryonic and perinatal lethal mouse mutant lines in order to investigate the causes of embryonic lethality. Here we show that individual whole-embryo RNA-seq of 73 mouse mutant lines (>1000 transcriptomes) identifies transcriptional events underlying embryonic lethality and associates previously uncharacterised genes with specific pathways and tissues. For example, our data suggest that *Hmgxb3* is involved in DNA-damage repair and cell-cycle regulation. Further, we separate embryonic delay signatures from mutant line-specific transcriptional changes by developing a baseline mRNA expression catalogue of wild-type mice during early embryogenesis (4–36 somites). Analysis of transcription outside coding sequence identifies deregulation of repetitive elements in *Morc2a* mutants and a gene involved in gene-specific splicing. Collectively, this work provides a large scale resource to further our understanding of early embryonic developmental disorders.

[1] Wellcome Sanger Institute, Wellcome Genome Campus, Cambridge CB10 1SA, UK. [2] Cambridge Institute of Therapeutic Immunology & Infectious Disease (CITIID), Jeffrey Cheah Biomedical Centre, University of Cambridge, Puddicombe Way, Cambridge CB2 0AW, UK. [3] European Molecular Biology Laboratory, European Bioinformatics Institute, Wellcome Genome Campus, Cambridge CB10 1SD, UK. [4] Division of Anatomy, MIC, Medical University of Vienna, Waehringerstr. 13, 1090 Wien, Austria. [5] The Babraham Institute, Babraham Research Campus, Cambridge CB22 3AT, UK. [6] Centre for Trophoblast Research, University of Cambridge, Downing Street, Cambridge CB2 3EG, UK. [7] Sir William Dunn School of Pathology, University of Oxford, Oxford OX1 3RE, UK. [8] The Francis Crick Institute, 1 Midland Road, London NW1 1AT, UK. [9]Present address: The Jackson Laboratory, 600 Main Street, Bar Harbor, ME 04609, USA. [10]Present address: Departments of Biochemistry & Molecular Biology and Medical Genetics, Cumming School of Medicine, University of Calgary, Calgary, AB T2N 4N1, Canada. [11]Present address: Camena Bioscience, The Science Village, Chesterford Research Park, Cambridge CB10 1XL, UK. [12]These authors contributed equally: John E. Collins, Richard J. White. Correspondence and requests for materials should be addressed to E.M.B.-N. (email: emb81@cam.ac.uk)

Animal models are a powerful surrogate for the study of human development and disease. Genetic screens have produced large mutation resources in the model organisms *Caenorhabditis elegans*[1–3], *Drosophila melanogaster*[4,5] and *Danio rerio*[6]. The creation of a collection of mutants covering every gene of the mammalian model *Mus Musculus* is currently underway, coordinated by the International Mouse Phenotyping Consortium[7] (IMPC). Adult knock-out mice undergo a range of systematic phenotypic assessments to define genotype−phenotype associations. These data provide vital information to further our understanding of human disease and developmental disorders[8].

It is estimated that around a third of all knock-out mutations in mice result in embryonic or perinatal (EP) lethality[9] and in these lines the adult phenotype can only be assessed in heterozygous individuals. Lethal lines, however, provide an opportunity to study embryonic anomalies and their correlates with human congenital disorders. The Deciphering the Mechanisms of Developmental Disorders (DMDD) programme was a 5-year project to systematically characterise EP lethal IMPC lines (defined as the absence of homozygous mutants after screening a minimum of 28 pups at P14).

The project has analysed over 240 mutant lines over the past 5 years and has created a public resource including embryonic high-resolution episcopic microscopy[10] (HREM) and placenta morphology assessment[11]. In this work, we present the transcriptomic analysis of a subset of these lines.

The DMDD programme assessed morphological phenotypes in E14.5 embryos, a time point when organogenesis is largely complete[12], and identified highly variable penetrance of morphological phenotypes[13,14]. Careful staging showed that the majority of homozygous embryos were delayed at E14.5[15]. Around 170 of the DMDD mutant lines (70%) did not produce viable homozygous embryos at E14.5 and therefore subsequent litters were analysed at mid-gestation (E9.5). In a third of these lines homozygous embryos could be retrieved at E9.5 with a subset displaying mild to severe developmental delay[16]. In addition to the embryos, placental tissues and yolk sacs were analysed which showed a high percentage of placental phenotypes among the embryonic-lethal mutant lines[17].

In this current work, we analysed transcriptome profiles of 73 embryonic-lethal mutant lines using individual whole embryos at E9.5. The embryos were staged using somite number to provide a more fine-grained and accurate developmental assessment than embryonic days post conception. Somite number connects stage to actual developmental progression independent from time post conception. Accurate staging is crucial to untangling the direct effects of the mutation on gene expression levels from gene expression changes due to mutant embryos being developmentally younger. However, because only the mutant embryos are delayed, it is difficult to separate these signals. We have therefore created a baseline of wild-type transcriptomes over the period of somite formation as a stage reference. Using this baseline we have developed a method to enrich for the direct effects of the mutation on gene expression and identify a separate signature caused by developmental delay. We have also produced a Shiny app called Baseline CompaRe (https://www.sanger.ac.uk/science/tools/dmdd/dmdd/) to enable researchers to visualise the data presented here and to apply the method of delay signal separation on other suitable datasets.

To perform these studies we use whole-organism RNA-seq since we and others have shown that this is a powerful method to define transcriptomic landscapes and identify genetic interactions[18–20]. Using this approach, we identified the molecular phenotypes of 53 homozygous mutant lines as well as of 20 lines where only heterozygous embryos could be retrieved at E9.5. We aimed to identify gene regulatory signatures underlying developmental delay and associate previously uncharacterised genes with specific pathways and developmental processes.

## Results

**The reference transcriptome of early mouse embryogenesis.** To produce a comprehensive baseline dataset of normal mRNA expression around E9.5, we collected somite-staged wild-type embryos from 4 to 28 somites (covering E8–E9.5) and 34 to 36 somites (E10.5 for comparison) with 3–4 embryos per somite number (Fig. 1a). In total, we produced RNA-seq libraries from polyA$^+$ mRNA for 111 individual embryos, which were sequenced at a mean read depth of 33.8 M mapped reads per sample (Supplementary Data 1). After mapping, we performed Markov clustering (MCL) on the data using the BioLayout *Express*$^{3D}$ software[21,22]. Clustering the samples using Pearson correlation produced three unconnected clusters, suggesting a batch effect (Supplementary Fig. 1a). Clustering the genes by correlation of expression levels to produce a gene expression network (Fig. 1b) confirmed that this was caused by a set of genes that had high expression for the 17–20 somite samples, which were all processed together (Fig. 1b, cluster labelled Batch outliers), while all other samples showed low expression. A large number of the genes in this cluster (41/194) were histone cluster genes, which are not polyadenylated[23] and so should not be present in poly-A pulldown RNA-seq. For these reasons, we believe this to be a technical artefact. Once these 194 genes (Supplementary Data 2) were removed, all of the samples clustered together in the sample correlation network (Supplementary Fig. 1a–b), showing that no large batch effects remain. We also removed Y chromosome genes and *Xist* (Fig. 1b), so that an imbalance in the sexes of sampled mutant and sibling embryos did not cause sex-specific genes to appear to be differentially expressed. Finally, genes encoded by the mitochondrial genome were removed as they are highly variable (Fig. 1b, MT genes) and so are prone to appearing differentially expressed by chance.

The clustered gene expression network also identified sets of genes with specific trajectories over the investigated timescale. The cluster labelled "Highly variable and decreasing" (Fig. 1b) contains genes whose expression is variable early on in the time course. The large range of expression levels at early time points means homozygous embryos could appear significantly different from siblings by chance at these stages. Other notable clusters included genes whose expression increases, decreases or is relatively stable (Fig. 1b, clusters labelled Increasing, Decreasing and Stable).

To investigate our ability to detect transcripts in whole embryos we compared our data with previously generated single-tissue RNA-seq[24] (six tissues dissected from E8.25 embryos). Of the protein-coding genes detectable with either whole-embryo or single-tissue RNA-seq (≥10 counts in any stage or tissue), the majority were found by both techniques (94.9%, 14,118 of 14,871; Supplementary Fig. 1c) and just a small subset were detected either in whole embryo only (2.1%, 306) or in single tissue alone (3.0%, 447; Supplementary Fig. 1d). In our transcriptome analysis of all embryonic samples we found a total of 346 previously unannotated mouse genes, of which 16 are protein coding and 330 are noncoding (Fig. 1c and Supplementary Data 3). These genes showed varied and dynamic expression patterns throughout our baseline (Fig. 1d). The baseline data have been incorporated in the Ensembl genome browser (https://www.ensembl.org/Mus_musculus/Info/Index) as RNA tracks which can be viewed by turning them on using the Control Panel (under the heading RNASeq models).

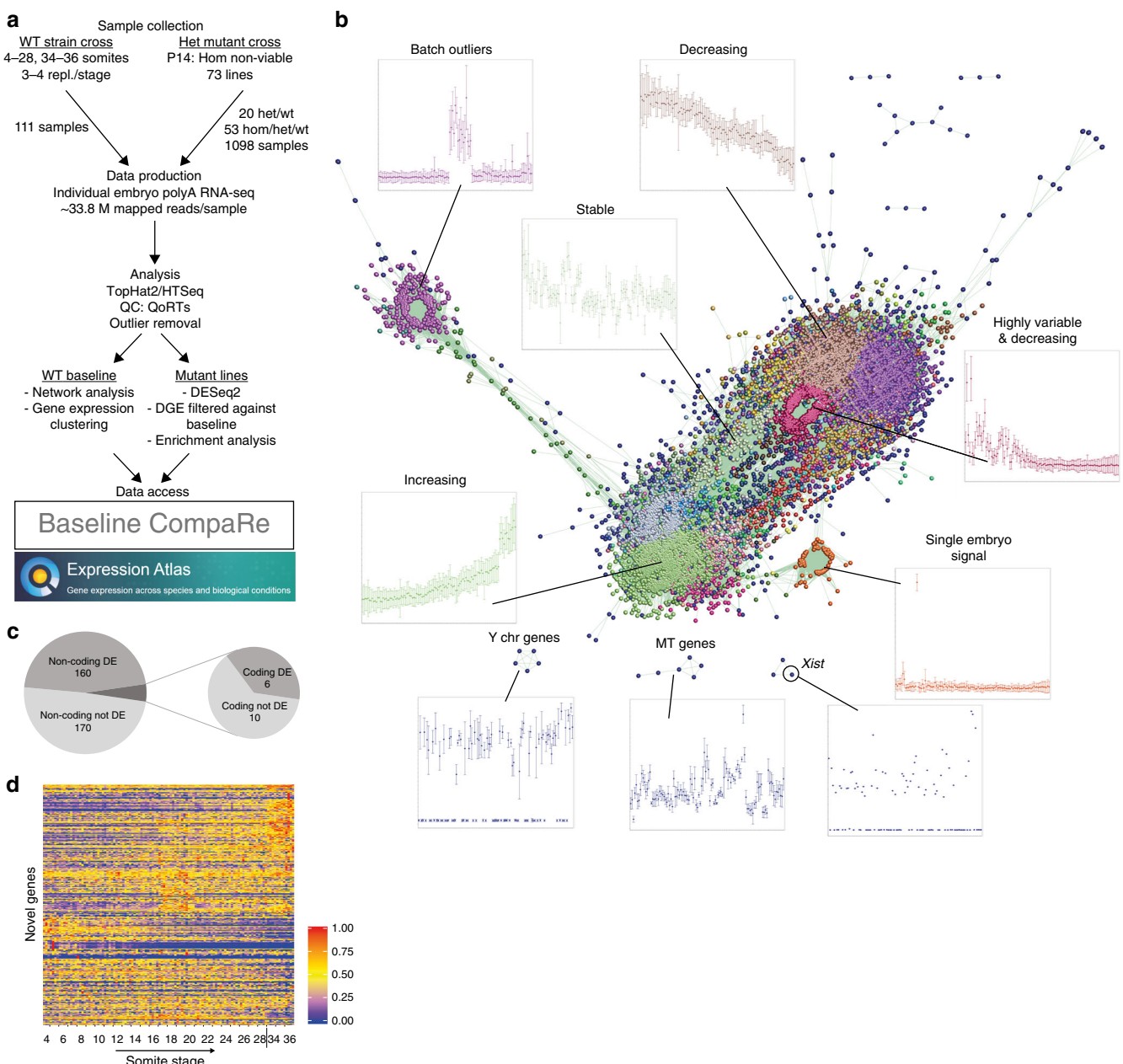

**Fig. 1** Experimental strategy and data processing. **a** Experimental workflow for E9.5 baseline and mutant line analysis. For the baseline, wild-type mice from heterozygous incrosses generated as part of the DMDD project were outcrossed (WT strain cross) and embryos from these crosses were collected. This was to produce embryos with a similar genetic background to the mutants. For the mutant lines, embryos were obtained from heterozygous intercrosses (Het mutant cross). The processed data can be viewed using Baseline CompaRe. The baseline data are available in Expression Atlas. **b** Clustered gene expression network identifies co-expressed genes across the baseline samples. The gene expression data are displayed using BioLayout Express[3D] as a network of genes (nodes) with edges connecting nodes whose expression values over all the samples have a Pearson correlation coefficient of ≥0.85. Nodes represent 9265 genes and are coloured according to Markov clustering (MCL). Surrounding graphs show expression of genes in a selection of clusters with individual embryos ordered by somite number on the *x*-axis, and gene expression in the cluster (unit variance scaled with standard deviation) on the *y*-axis. Batch outliers (top left) are genes that were blacklisted from further analysis. Decreasing/increasing genes are the ones most indicative of stage as they change the most across the time course, whereas the cluster labelled Stable is more consistent. Highly variable and decreasing is a set of genes that have high variance even among embryos at the same stage. Single embryo signal is a cluster of genes that are expressed at high levels in just one embryo. *Xist*, Y chr genes and MT genes (genes encoded by the mitochondrial chromosome) were removed due to high variability. **c** Venn diagram of the categories of novel (previously unannotated) genes identified using all RNA-seq data across all embryos. **d** Heatmap of expression profiles of the novel genes. Each row is a gene and each column a sample in somite number order. In each row, expression data are scaled to the maximum value (set to 1) for that row. The genes are ordered by hierarchical clustering. Source data are provided as a Source Data file

**Dissection of transcriptional profiles of 73 mutant lines**. We collected embryos at E9.5 from heterozygous incrosses of homozygous lethal lines (Fig. 1a, Het mutant cross). Each embryo was staged according to somite number, genotyped, and processed for mRNA extraction (Supplementary Data 4). Genotyping showed that we obtained between 2 and 16 homozygous individuals for 53 lines, whereas for the remaining 20 lines only heterozygous and wild-type embryos could be recovered at this developmental stage. Embryonic lethality is often preceded by general developmental delay, which we observed in 31.6% of homozygous embryos (79 of 250; Supplementary Data 4). In these cases, we reasoned that the transcriptional response due to the mutation would be confounded with a secondary, more generic, delay signature that could be visualised by principal component analysis (PCA). Principal component (PC) 3 (by definition the third strongest signal in the data, accounting for 6.3% of all the variance) correlates very well (Pearson correlation coefficient = −0.865) with the recorded somite number (Fig. 2a). The genes that contribute most to this PC are the ones that change the most (either increasing or decreasing) across the time course and are therefore indicative of stage (Fig. 1b, clusters labelled Increasing and Decreasing; see gene list in Supplementary Data 5). These included well-known developmental transcription factors such as *NeuroD1*, *Cdx2* and *Myog*, suggesting that PC3 represents a common developmental stage signal likely to affect the differential expression analysis.

**Accounting for developmental delay**. We therefore devised a strategy to separate the transcriptional signal due to delay from the primary transcriptional response to gene disruption. Because embryos at E9.5 have 21–29 somites[25], we defined a mutant as developmentally delayed if at least one of the embryos had fewer than 20 somites, allowing for a one-somite counting error. We subjected transcriptional profiles for mutant lines defined in this way to differential gene expression (DGE) analysis in three ways (Fig. 2b, Supplementary Fig. 2). First, we identified DGE by running homozygous samples against their siblings (Supplementary Fig. 2a, median 4 vs. 9 embryos). Second, we included baseline samples for the stages present in the experimental embryos to better estimate normal variance for each gene (Supplementary Fig. 2b). For example, in the *Brd2* mutant line, the experimental samples are from 13, 15, 19–20, 22 and 25–28 somite stages so baseline samples from the same stages were included (Supplementary Fig. 2b, grey shading). Third, in addition to including the baseline as above, we defined the number of somites as a group in the DESeq2 model to take developmental stage for each embryo into account (Supplementary Fig. 2c). Overlapping the resulting DGE lists allowed us to separate four DGE profiles (Fig. 2c). (1) Mutant Response, defined as DGE outside of sibling and baseline expression range. (2) Delay, where mutant gene expression is different from wild-type and heterozygous siblings, but the same as in somite-matched baseline embryos and therefore likely due to developmental delay. (3) No Delay, where mutant gene expression is the same as in siblings and therefore not affected by the morphological delay. (4) Discard, where DGE is unreliable due to normally high variance in the baseline or overall low expression.

The effect of this filtering is to make the signal of the specific mutant response stronger (Fig. 2d, e). For example, *Fcho2* is a gene required for the production of clathrin-coated vesicles[26]. In the analysis of Gene Ontology (GO) terms for the *Fcho2* mutant line, in which homozygous embryos are delayed at E9.5, terms such as intracellular protein transport, endosomal transport and asymmetric protein localisation are promoted to the top of the list when using the Mutant Response DGE list only (Fig. 2d).

Similarly, the transcriptome profile in mutants of *Hira*, a histone chaperone that also affects histone gene regulation[27], is enriched for genes associated with GO terms such as histone exchange, nucleosome assembly, regulation of transcription and mRNA splicing, after taking delay into account (Fig. 2e). We have developed an online tool called Baseline CompaRe (https://www.sanger.ac.uk/science/tools/dmdd/dmdd/) that enables researchers to separate developmental delay from primary transcriptional response in suitable RNA-seq datasets. Users can upload count data and the app will run DESeq2, overlap the results, and produce DGE lists for the four response categories.

**The same tissues are affected by delay across mutant lines**. Given that there is a set of genes prone to appearing in DGE lists due to mutants being developmentally delayed, we tested whether the same genes appear in the Delay category for multiple mutants. Generally speaking, the overlap between genes in the Delay category from different mutant lines is small (Fig. 3a). Of the 73 mutant lines, 40 were designated as delayed and homozygous embryos could be recovered in 32. For 6 of the 32 lines, no genes were assigned to the Delay category. In the remaining 26 lines, the largest pairwise overlap of the DGE lists is 30.9% (Source Data for Fig. 3a). The largest overlaps generally appear in the most delayed lines which tend to have the longest Delay category lists.

We then sought to investigate whether the same tissues, rather than individual genes, are affected by delay in different mutant lines. To this end we analysed enrichment of Edinburgh Mouse Atlas Project Anatomy[28,29] (EMAPA) ontology terms associated with the gene lists (Fig. 3b–e). This confirmed that 17 of 26 delayed lines shared tissue enrichments in the Delay category (Fig. 3c) whereas the Mutant Response DGE lists produced diverse enrichments (Fig. 3b). As expected only a few No Delay and Discard lists produced enrichments, with no common theme among mutant lines (Fig. 3d, e). Of the genes that appeared in the No Delay lists frequently (defined as being in more than six mutant lines), those involved in cardiovascular development, such as *S1pr1* and *Pecam1*, and nervous system genes, for example *Gbx2* and *Wnt5b*, are enriched (Supplementary Fig. 3, Supplementary Data 6). This suggests these tissues can develop normally despite apparent global developmental delay.

**Transcriptional similarities of ciliopathy mutants**. Half of the lines analysed here had at least one embryo with fewer than 20 somites (Fig. 4a, coloured bars in the 12–14 Theiler Stage categories). Notably, this did not only occur in homozygous embryos, reflecting variation in the pace of development even in wild-type embryos. In most lines (53 of 73), the targeted gene was at significantly lower levels in either homozygous or heterozygous mutants than in wild-type embryos (Fig. 4b, see Source Data for Fig. 4b for *p* values). Applying the delay analysis described above (Accounting for Developmental Delay) substantially reduced the Mutant Response category DGE lists for the most delayed mutant lines. Twenty-two of the delayed lines had a shortened DGE list with a mean reduction of 47% (Fig. 4c; comparing pre- and post-filter columns). In a few cases (e.g. *Tial1*) the lists are longer, most likely due to an increase in detection power caused by inclusion of the baseline samples. Generally, there was no, or only a very small, transcriptional response when comparing expression between heterozygous and wild-type embryos (Fig. 4c, het vs. wt column). There was no correlation between severity of phenotype and timing or level of expression of the mutated gene during the baseline stages (Fig. 4d). We assessed the medical relevance of this dataset and found that over a third (28/73) of the genes included in this study have human orthologues that have previously been implicated in human disease (Fig. 4e).

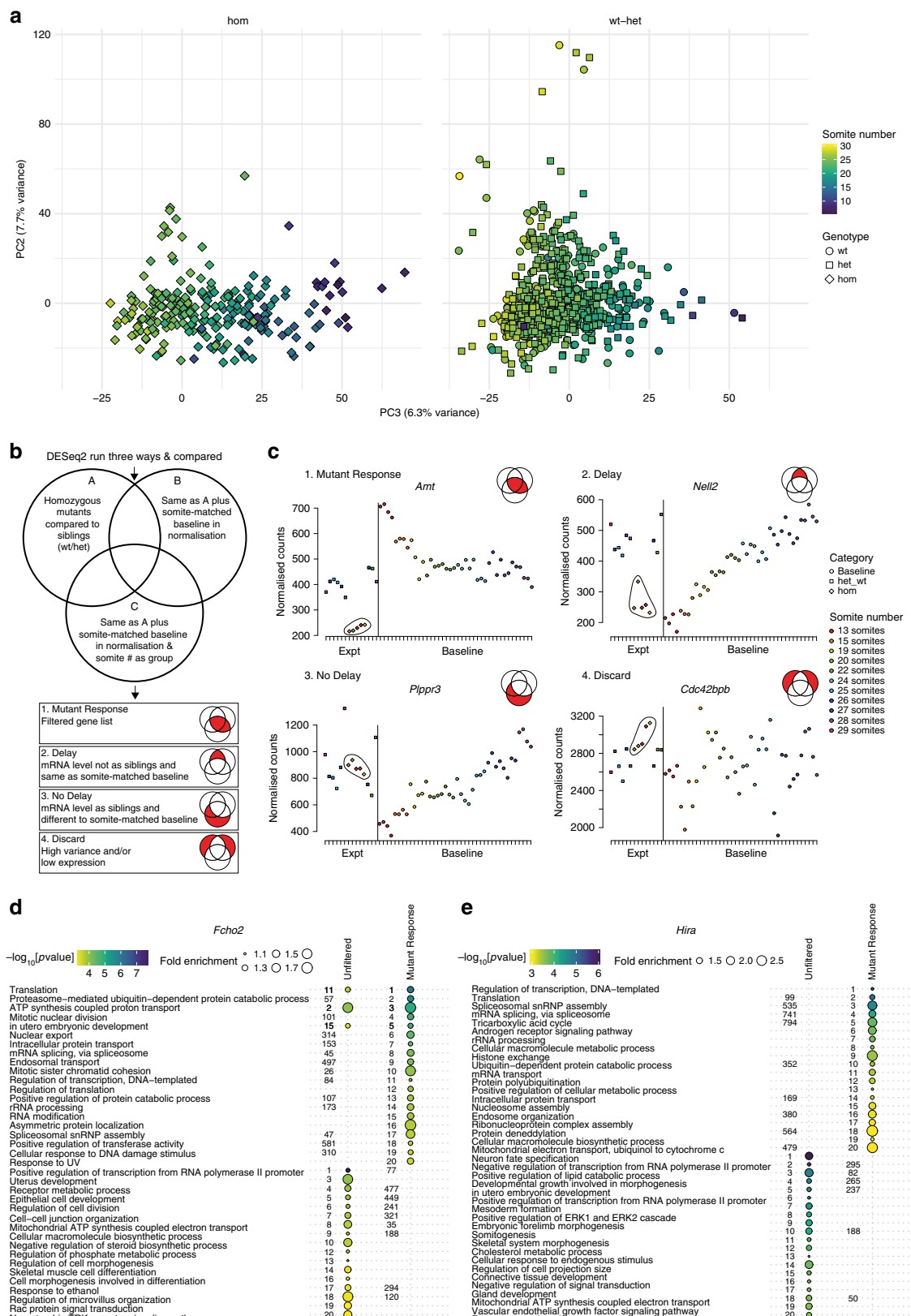

For a comparative assessment of the mutant responses across all lines, we performed enrichment analysis for GO terms and EMAPA terms. In contrast to the Delay response, enriched GO/EMAPA term sets for the Mutant Response DGE lists differed from line to line (Figs. 3c, 5a). The most frequently enriched GO term across all lines was "response to

hypoxia" (GO:0001666) which might reflect stress or placental defects[17].

Although generally the enriched terms differed between lines, small groups of lines that shared similar sets of enriched terms were found (Fig. 5a). One group of mutants in which similar sets of terms were enriched contained genes previously implicated in

**Fig. 2** Separating the delay signal. **a** Principal component analysis (PCA) of all experimental samples in the study using the data for all genes. Principal component (PC) 2 and PC3 are plotted, points are coloured by recorded number of somites. The variance explained by each component is in brackets. The left-hand plot shows homozygous embryos (diamonds) and the right-hand heterozygous (squares) and wild-type (circles) embryos. Somite number is correlated with PC3. **b** Analysis strategy. Each delayed line was analysed with DESeq2 in three different ways (see Supplementary Fig. 2) and the differentially expressed (DE) gene lists overlapped. Four categories of genes were produced based on their position in the Venn diagram; 1. Mutant Response, 2. Delay (mRNA abnormal), 3. No Delay (mRNA as wt) and 4. Discard. **c** Examples of genes from each category. Plots show expression levels (normalised counts) of the gene in embryos (siblings = squares, homozygous = diamonds) from the *Brd2* mutant line (Expt = experimental samples) as well as baseline samples (circles) of matching somite stages. Homozygous mutants are circled. Genes shown are Mutant Response: *Amt* (ENSMUSG00000032607), Delay: *Nell2* (ENSMUSG00000022454), No Delay: *Plppr3* (ENSMUSG00000035835) and Discard: *Cdc42bpb* (ENSMUSG00000021279). **d**, **e** Effect of filtering on Gene Ontology (GO) enrichment. Plots of the top 20 (by *p* value) enriched GO terms (Biological Process) in both unfiltered and Mutant Response DE gene lists. The size of the points represents fold enrichment of the term (Observed/Expected) and they are coloured by $-\log_{10}[p$ value] (no point means the term is not in the top 20 for that gene list). The numbers are the position of the term in each list ranked by *p* value (no number means the term is not enriched for that gene list). **d** Enrichments for the *Fcho2* (ENSMUSG00000041685) mutant line. **e** Enrichments for the *Hira* (ENSMUSG00000022702) mutant. Source data are provided as a Source Data file

ciliopathies (Fig. 5a, grey box). Impaired morphology or function of cilia very early in development can lead to a disruption in left-right asymmetry which is easily defined by *situs inversus* phenotypes. Another characteristic early ciliopathy phenotype is polydactyly, which is caused by decreased Shh signalling in the developing limb bud. Among the mutant lines in this study we found five genes which have been previously linked to human ciliopathies: *B9d2*, *Cc2d2a*, *Rpgrip1l*, *Ift140* and *Nek9*[30]. The GO/ EMAPA enrichment profiles of these mutant lines clustered together with a sixth gene, the kinesin 2 complex member Kifap3, which was not previously implicated in ciliopathies (Figs. 3b, 5a). This prompted us to investigate the profile of another complex member, *Kif3b*. Both genes show ciliopathy phenotypes in mice, namely *situs inversus* in *Kif3b* homozygous embryos[31] and polydactyly in *Kifap3* homozygous embryos (IMPC[16]). These early developmental phenotypes are caused by impaired cilia-dependent Shh signalling[32,33].

Examining the overlap of DGE in the seven lines showed that the majority of genes that were differentially expressed in at least four of the mutants (Fig. 5b) are known downstream targets of Shh signalling[34–43] or are implicated in interactions with the Shh pathway[44–49]. The remaining three genes in this list are two poorly described noncoding RNAs (*Gm3764*, *Gm38103*) which might be potential Shh interactors and the transcription factor *Sox21* which has not been associated with Shh signalling before. *Gm3764*, like *Shh*, is expressed in the embryonic floor plate at E8.5 (Expression Atlas[50]). *Gm38103* is located on chromosome 12 between *Nkx2-1* and *Nkx2-9*, which are also downstream of Shh signalling; however, this noncoding RNA could also constitute an alternative 3′ end of *Nkx2-9*.

**Assigning function to known and previously uncharacterised genes.** After establishing that our approach can identify specific transcriptome profiles from whole-organism RNA-seq in known pathways like Shh signalling in ciliopathy genes, we tested if we could link less well-described genes to known molecular or developmental pathways. We analysed the DGE list for the *Zkscan17* mutant line by GO term enrichment, which produced terms that could be grouped into central nervous system, cardiac and microtubule development or functions. The human homologue *ZNF496* has been shown to bind to *JARID2*, suggesting a role in heart development[51]. The differentially expressed (DE) genes responsible for the GO term annotations are mainly associated with neuronal function (Fig. 5c). For example, transcripts of three genes coding for synapse-located *Slitrk* proteins[52], whose human homologues are associated with schizophrenia and Tourette syndrome[53], are less abundant in *Zkscan17* mutants. This implicates Zkscan17 in neuronal function, which has not been defined before.

Another example where we can suggest a possible function for a previously uncharacterised gene is *Hmgxb3*. The DE genes in this line are associated with just 23 GO terms that are mainly linked to DNA damage/repair and G1/S-phase transition, but also cilium morphogenesis (Fig. 5d). Because some of the cilia genes are also components of centrioles, our work suggests they might be involved in the cell cycle instead of, or as well as, ciliogenesis.

Nadk2 catalyses the phosphorylation of nicotinamide adenine dinucleotide (NAD) to NADP in mitochondria. The allele used in this study, *Nadk2*^tm1b(EUCOMM)Wtsi, shows complete penetrance of pre-weaning lethality and moderate to slight morphological delay at E9.5[16] (Fig. 4a), whereas *Nadk2*^tm1a(EUCOMM)Wtsi produces phenotypic homozygous adults[54]. We found 5976 DE genes in the transcriptional mutant response of the former. One of the most prominent signals was a marked reduction in expression of heme biosynthesis enzymes and globins (Fig. 6a–c). Expression of all but one of the enzymes in the heme pathway was reduced. The mitochondrial iron importer *Slc25a37* and all globin genes also showed large decreases in expression demonstrating a link between reduced or absent heme and a dramatic down-regulation of globins. Although baseline globin gene expression increases from E8 to E10.5, we were able to distinguish the mutant response from delay—due to the magnitude of the transcript reduction (Fig. 6c).

We detected reduced expression of a number of erythrocyte and blood group genes, such as *Slc4a1*, *Gypa* and *Kel* (Fig. 6a). Taken together, these data suggest a substantial reduction or absence of erythrocytes in the homozygous embryos. To confirm this, we analysed HREM data of six *Nadk2*^tm1b(EUCOMM)Wtsi homozygous mutants and three wild-type littermates (Fig. 6d–i). Homozygous mutants showed either global delay or delayed development of the caudal body parts (Fig. 6e, f) and malformations of pharyngeal arch arteries and dorsal aortae. Neither heart, arteries nor veins contained erythrocytes. Only in some vessel segments could scattered blood cells be detected (Fig. 6h, i). By contrast, blood was readily detected in wild-type littermates (Fig. 6g). This demonstrates a lack of erythrocytes in the homozygous mutant embryos, as suggested by the transcriptome profiling.

**Transcription analysis outside gene annotation.** During the analysis, we noticed abnormal expression and thus potential de-repression of repeats (as defined by RepeatMasker) in 12 of the 73 expression profiles (Supplementary Fig. 4a). In light of increasing evidence of the importance of repeat elements and their control in processes such as gene regulation[55], germline differentiation[56] and cancer[57], we examined transcription outside annotated exons (Fig. 7a). After identifying the differentially expressed repeat instances (DERs), for example, long interspersed elements (LINEs), we found that the *Dhx35* and *Morc2a* mutant lines

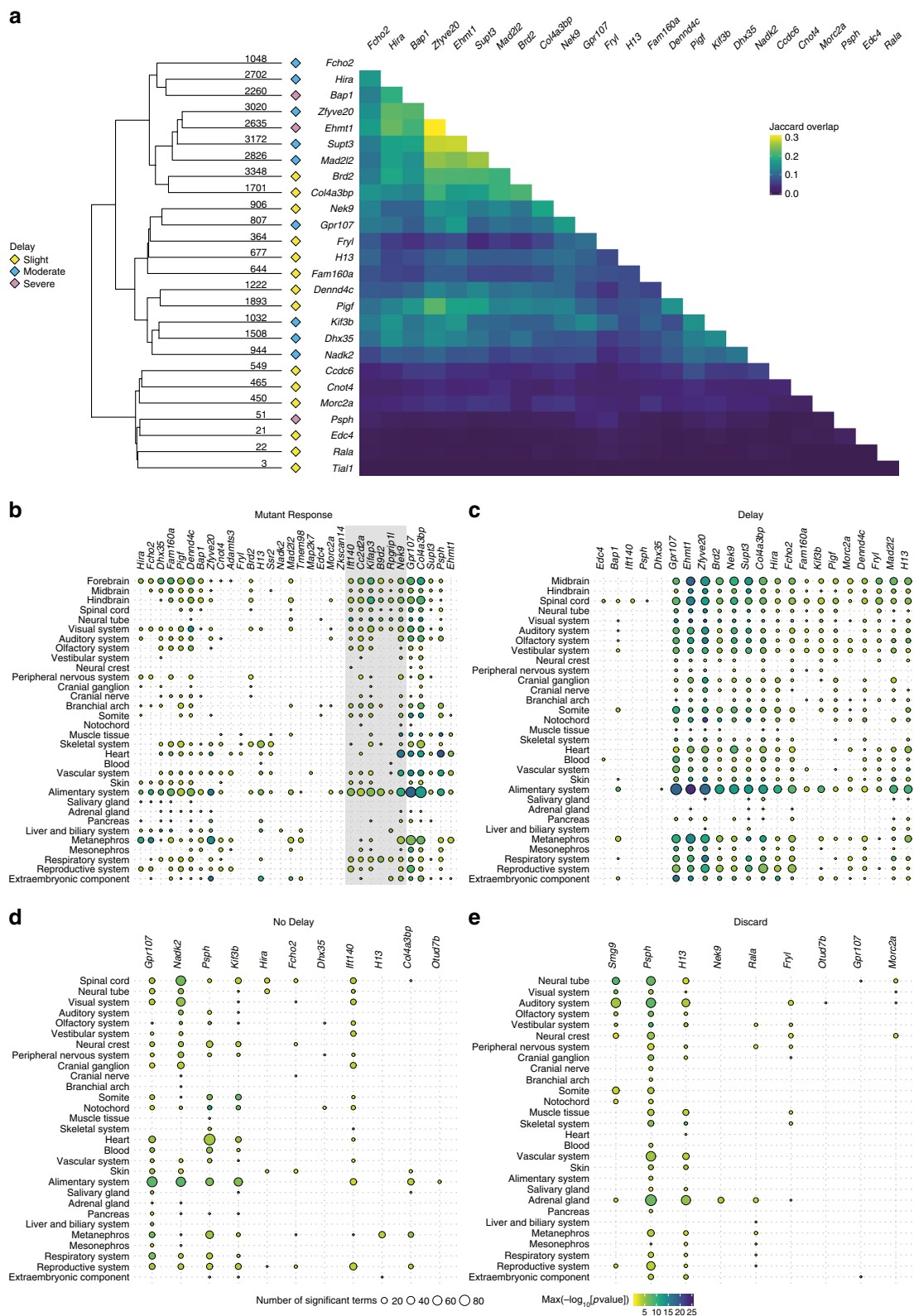

showed large numbers of DERs in specific genomic region types (Fig. 7b). In contrast to the other 11 lines, DERs in the *Dhx35* line were mostly intronic whereas those in *Morc2a* were mostly intergenic (Fig. 7b).

**Intronic DERs in *Dhx35* mutants.** *Dhx35* encodes a putative RNA helicase suggesting involvement in RNA structure

regulation. DERs in *Dhx35* mutants were mostly intronic and dominated by SINEs in the sense orientation (Fig. 7b, Supplementary Fig. 4b). The vast majority of intronic DERs in *Dhx35* mutants were not found in the other mutants (Supplementary Fig. 4c). One hundred and thirty-nine genes (115 unique to *Dhx35*) contained a significant enrichment of 407 intronic DERs when compared to the total number of intronic repeats (Fig. 7c;

**Fig. 3** Anatomical term enrichment analysis. **a** Pairwise overlaps between Delay gene lists. The heatmap on the right shows the Jaccard similarity index (number in both lists/number in either list) for pairs of delayed mutant lines. The heatmap has been hierarchically clustered and the tree is displayed on the left along with the number of genes in each Delay gene list. The category of delay for each line is indicated by coloured diamonds (yellow = Slight, blue = Moderate, purple = Severe). **b–e** Bubble plots of the enriched Edinburgh Mouse Atlas Project Anatomy (EMAPA) terms produced from the four categories of differentially expressed genes. The mutant lines are displayed on the x-axis and the terms are on the y-axis. The ordering of mutants on the x-axis was determined by hierarchical clustering of the overlap (Jaccard Index) of terms between lines. The enriched terms on the y-axis were simplified by aggregating to terms at the top of the EMAPA ontology graph. The size of the bubbles represents the number of terms that have been simplified to the higher-level term and they are coloured by maximum $-\log_{10}[p$ value]. Mutant lines with no enrichments have been excluded from the plots. **b** Mutant Response. Small groups of lines have similar tissues enriched. The grey box highlights the ciliopathy lines that cluster together based on similarity of enriched EMAPA terms. **c** Delay. The enriched tissues are more uniform across the lines. **d** No Delay. **e** Discard. For No Delay and Discard fewer lines have enriched terms and there is little pattern to the enriched tissues

for $p$ values see Source Data for Fig. 7c). Expression of all but one of the enriched intronic DERs was increased in *Dhx35* mutants compared to wild-type embryos (Supplementary Fig. 4d). One explanation for this could be that the genes surrounding the DERs are differentially expressed, causing the repeats they contain to appear differentially abundant as well. If this were the case, the fold change of the gene and of the DERs it contains should be correlated. However, the mean fold change of DERs unique to *Dhx35* does not correlate with the fold change of the surrounding gene, whereas it does for DERs that occur in other mutant lines in addition to *Dhx35* (Fig. 7c; Pearson correlation coefficient = 0.905). This suggests the differential expression in DERs unique to *Dhx35* is not caused by differential expression of the genes harbouring the repeats.

Individual inspection of introns containing DERs suggested an increase in intron retention in *Dhx35* rather than de-repression of specific repeats (Supplementary Fig. 5). We therefore used iDiffIR (https://bitbucket.org/comp_bio/idiffir), a tool for assessing intronic and exonic read depths, to check for increased intron retention in *Dhx35* (and *Morc2a* and *Hmgxb3* as controls). A similarly small proportion of introns showed significant intron retention—for *Dhx35*, 73 of 39,203 introns tested were significant (adjusted $p$ value < 0.05 after multiple testing correction), and for the *Hmgxb3* and *Morc2a* controls 58 (of 37,971) and 51 (of 36,807) introns, respectively, showed significant retention (iDiffIR results files can be downloaded at https://doi.org/10.6084/m9.figshare.7613528). This suggests that intron retention is not occurring genome-wide, but rather that *Dhx35* is either acting on specific introns post-transcriptionally or specific regions in the introns are transcribed independently of the surrounding gene.

**De-repression of intergenic L1 LINEs in *Morc2a* mutants.** *MORC2*, the human orthologue of *Morc2a*, has been linked to Charcot−Marie−Tooth disease and repression of hetero-chromatin associated with the HUSH complex[58]. Furthermore, MORC2 and HUSH complex subunits can bind and silence young LINE-1s in cell lines[59]. In contrast to the other 12 mutant lines, 80% (1028 of 1293) of the DERs in *Morc2a* were intergenic (Fig. 7b). Among instances within introns, we found DE LINEs— and, at a lower frequency, DNA and LTR families—were more often anti-sense than sense, unlike almost all other repeat groups in all mutants, which were more likely to be sense (Supplementary Fig. 4b). This suggests the repeat expression is independent of the gene. We performed enrichment analysis for each repeat family (Fig. 7d) and identified 14 families enriched for DERs (adjusted $p$ value < 0.05, binomial test, see Source Data for Fig. 7d). The majority of these were from the LINE and LTR groups, all of which showed increased expression (Fig. 7d, heatmaps). The L1MdGf_I family, a young and expanding subfamily of L1 elements potentially capable of retrotransposition[60], stands out, with 693 DERs out of 3132 instances. The majority (86%) of

these are intergenic with only three (0.4%) found in exons, suggesting a preference for transcriptionally silent regions. These L1MdGf_I repeat instances are rarely expressed in sibling (Fig. 7d, heatmaps) or baseline embryos (Supplementary Data 7). The enrichments for L1MdGf_I, MMERGLN-int and MMETn-int families were preserved after filtering for repeat lengths greater than 4 kbp, suggesting it is not an artefact of not having full-length repeats (Fig. 7d, Supplementary Fig. 4e). Notably, one MMERGLN-int and 57 (8.2 %) L1MdGf_I DERs map to the Y chromosome, but are found in RNA derived from both male and female embryos (Fig. 7d, heatmap; see Source Data for Fig. 7d2). This raises the possibility that these repeat instances have undergone transposition, are present as copies on autosomes, and are thus capable of generating mRNA in female embryos. To look at the repeat families independently of genome location we took the total number of reads which map to each repeat family, calculated the mutant versus sibling $\log_2$ fold change and plotted this against the total number of counts (Supplementary Fig. 4f). This confirms that overall L1MdGf_I, MMERGLN-int and MMETn-int families increase in abundance in the *Morc2a* mutant embryos.

We also examined the list of 1392 DE genes associated with the *Morc2a* mutant using reads mapped independently of the gene annotation, and found a number of genes of interest. *Tug1*, a non-coding anti-sense RNA of *Morc2a*, and *Cbx7*, were both increased in expression—these are known to be involved in polycomb-mediated repression and cell-cycle arrest[61]. Other cell-cycle genes were also upregulated, including *Ccng1*, *Cdkn1a* and *Gtse1*. *D1Pas1* [62] and *Zp3r*, which despite being described as testis specific were expressed in both male and female homozygous embryos. Taken together, these dysregulated genes suggest that male germline or gonad development and the cell cycle may be affected in *Morc2a* mutants.

## Discussion

The DMDD programme has analysed 247 embryonic or perinatal lethal mutant lines over the last 5 years and created an important public resource[11]. In this study, we have generated transcriptional profiles for 73 of these lines to identify the molecular signatures underlying embryonic lethality. We have developed a procedure to enrich for primary responses to gene loss by separating out the transcriptional changes due to developmental delay. This has enabled us to assign function to previously poorly characterised genes. The results are available to download (https://doi.org/10.6084/m9.figshare.c.4127441) and can be visualised and interrogated in our online tool Baseline CompaRe.

For 20 of the lines only heterozygous and wild-type embryos could be retrieved at mid-gestation. We detected only very limited gene expression changes in the heterozygotes even though the transcripts of the targeted genes themselves were less abundant. By contrast, adult phenotypes have been observed in hetero-zygotes in 12 of these lines, for example, decreased bone density

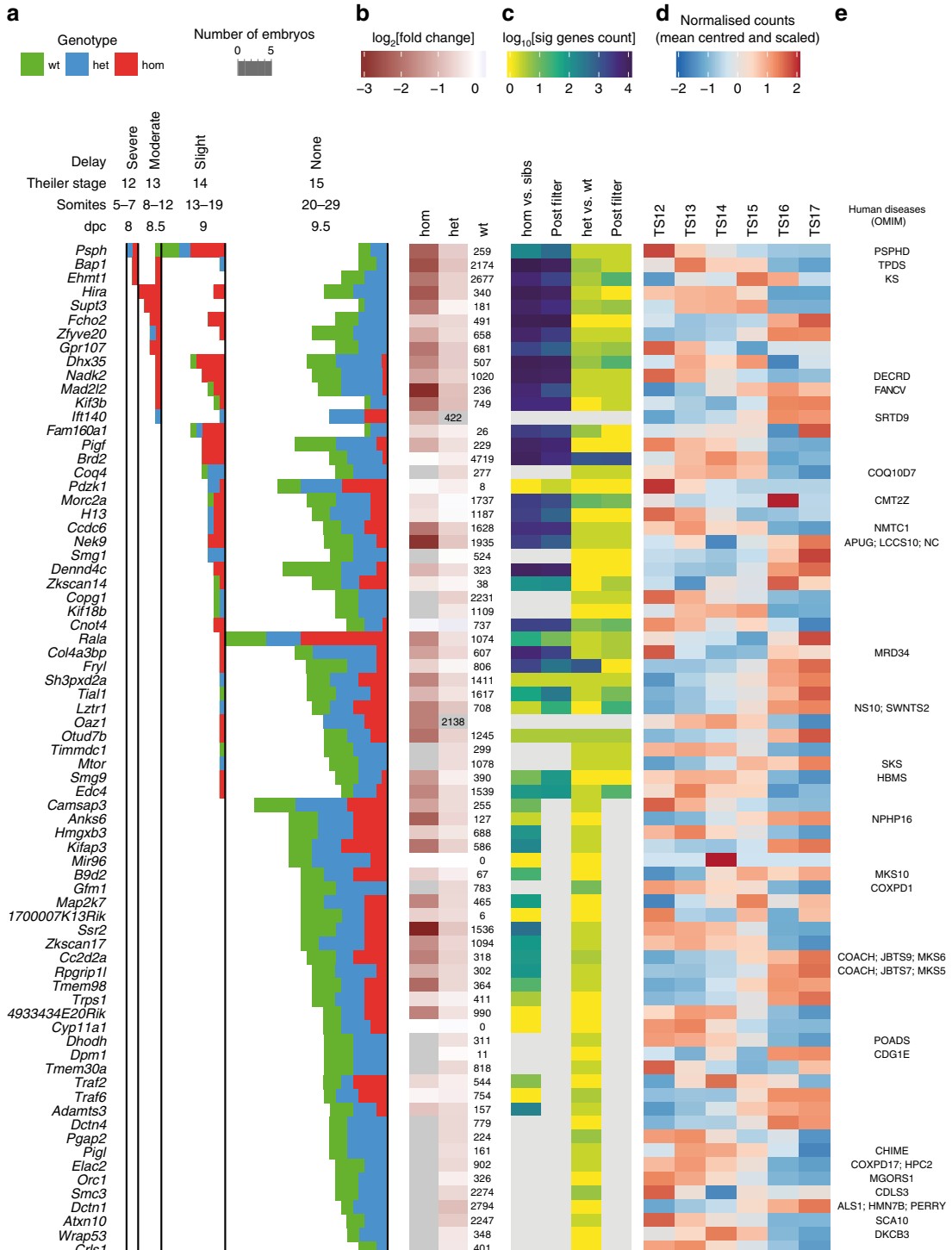

**Fig. 4** Summary of mutant lines. **a** Diagram of the stages of embryos collected coloured by genotype (green = wild type, blue = heterozygous, red = homozygous). The horizontal length of the bars represents the number of embryos at each Theiler stage for each genotype. The mutant lines are arranged from top to bottom by decreasing amount of delay. Lines are categorised by the most delayed embryos collected (TS12 = Severe, TS13 = Moderate, TS14 = Slight, TS15 = None). For two of the lines (*Ift140* and *Oaz1*), homozygous and heterozygous embryos were collected, but no wild-type siblings were present. **b** Heatmap of expression relative to wild-type (log$_2$[fold change]) of the targeted gene itself (hom = homozygous vs. wild type, het = heterozygous vs. wild type). The numbers in the wt column are mean normalised counts in the wild-type embryos for comparison (except for *Ift140* and *Oaz1* where the mean counts are for the het embryos). **c** Heatmap of the number of genes called as significantly differentially expressed (DE; log$_{10}$ scaled). hom vs. sibs = homozygous embryos compared to siblings (heterozygous and wild-type), het vs. wt = heterozygous compared to wild-type embryos. Columns labelled post-filter show the numbers of DE genes after the delay analysis was applied for delayed lines. Grey boxes are where no comparison was done, for example, where no homozygous embryos were recovered or the delay analysis was not applied because the line was not delayed. **d** Expression of the targeted gene in the wild-type baseline averaged by Theiler stage and displayed as mean centred and scaled normalised counts. **e** Human syndromes (Online Mendelian Inheritance in Man) associated with the targeted gene. Source data are provided as a Source Data file

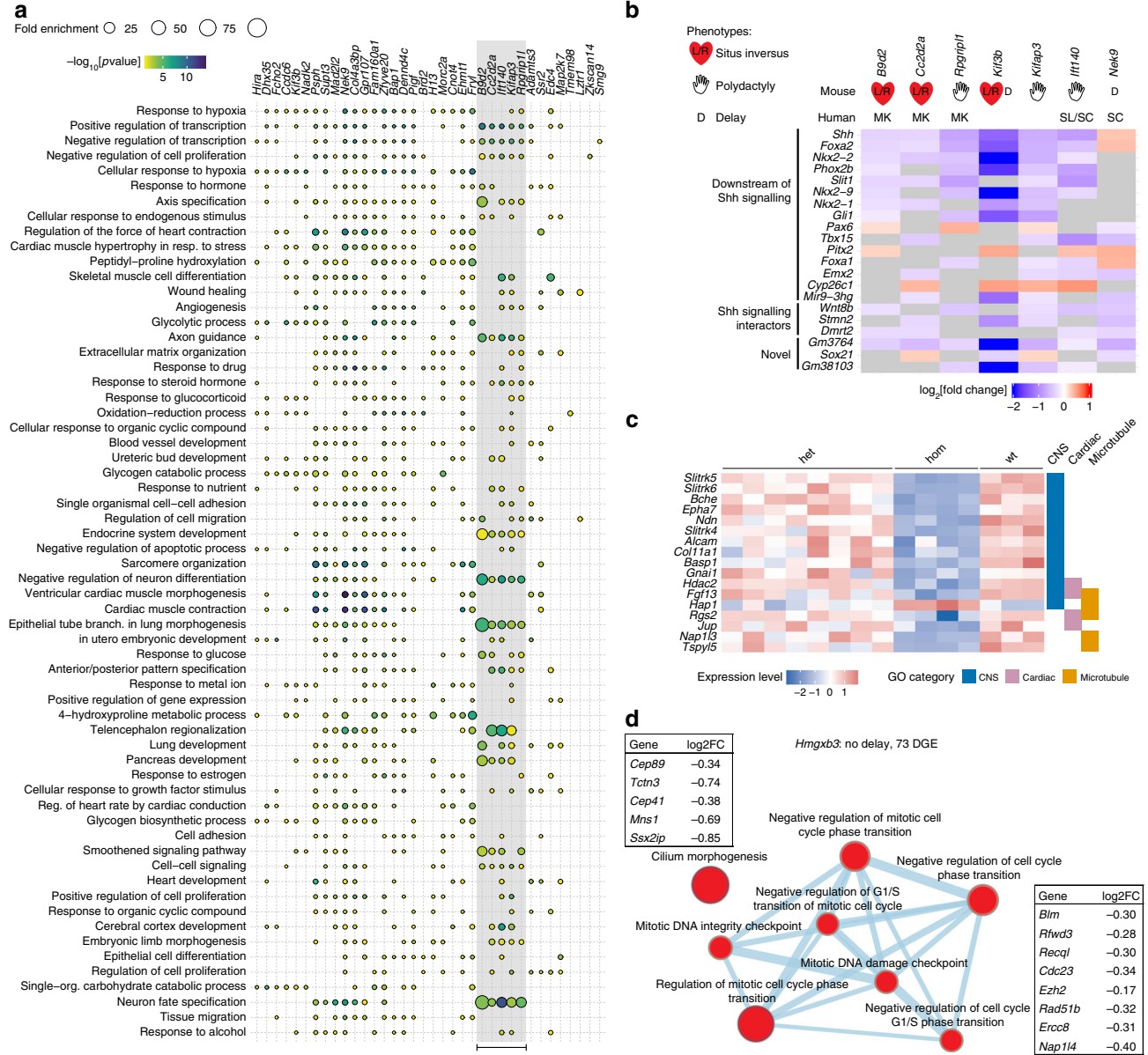

**Fig. 5** Mutant response overview and examples. **a** Bubble plot of the enriched Gene Ontology (GO) terms shared across most mutant lines, with lines on the *x*-axis and GO terms on the *y*-axis. The ordering of mutants on the *x*-axis was determined by hierarchical clustering of the overlap (Jaccard Index) of terms between lines. The size of the bubbles represents fold enrichment (Observed Genes/Expected) and they are coloured by $-\log_{10}[p$ value]. The group of ciliopathy mutants are highlighted with a grey box and a bar at the bottom. **b** Heatmap of the $\log_2$[fold change] of genes that are differentially expressed (DE) in at least four of the seven mutants identified as having similar ciliopathy profiles. Mutant lines are shown on the *x*-axis and DE genes on the *y*-axis. Phenotypes associated with mutations in human and mouse are shown above the heatmap. The DE genes have been categorised into three groups, Downstream of Shh signalling, Shh signalling interactors and Novel. **c** Heatmap of 17 DE genes from the *Zkscan17* mutant line associated with central nervous system (CNS), cardiac or microtubule GO terms. The heatmap displays expression values as mean centred and scaled normalised counts in all samples and the GO categories associated with each gene are shown to the right of the heatmap. **d** Network diagram produced by Enrichment Map (Cytoscape App) of the GO term enrichment in the *Hmgxb3* mutant line. The nodes represent enriched GO terms and the edge widths are proportional to the overlap of genes annotated to each term. Not all enriched GO terms are included. Source data are provided as a Source Data file

in *Atxn10* (IMPC[16]). This suggests that these defects arise at stages later than E9.5.

Many homozygous embryos were already developmentally delayed at E9.5. Indeed, the PCA analysis demonstrated that the third largest source of variation (PC3) in the experimental embryos can be attributed to developmental progression as measured by somite number (Fig. 2a). We therefore devised a strategy to isolate the direct transcriptional response to gene disruption from a more general delay signal in the transcriptome profiles. Comparing GO term enrichments with and without

filtering showed that GO terms matching the previously established functions of well-characterised genes like *Fcho2* and *Hira* (Fig. 2d, e) moved to the top of their enrichment lists after filtering, demonstrating the validity of this approach. It is important to note that, using this system, we may miss some genes that are directly affected by the mutation, but also change substantially in expression level over the time period of the baseline.

We have used somite number to stage embryos and it is possible that a mutation directly affects somite number without

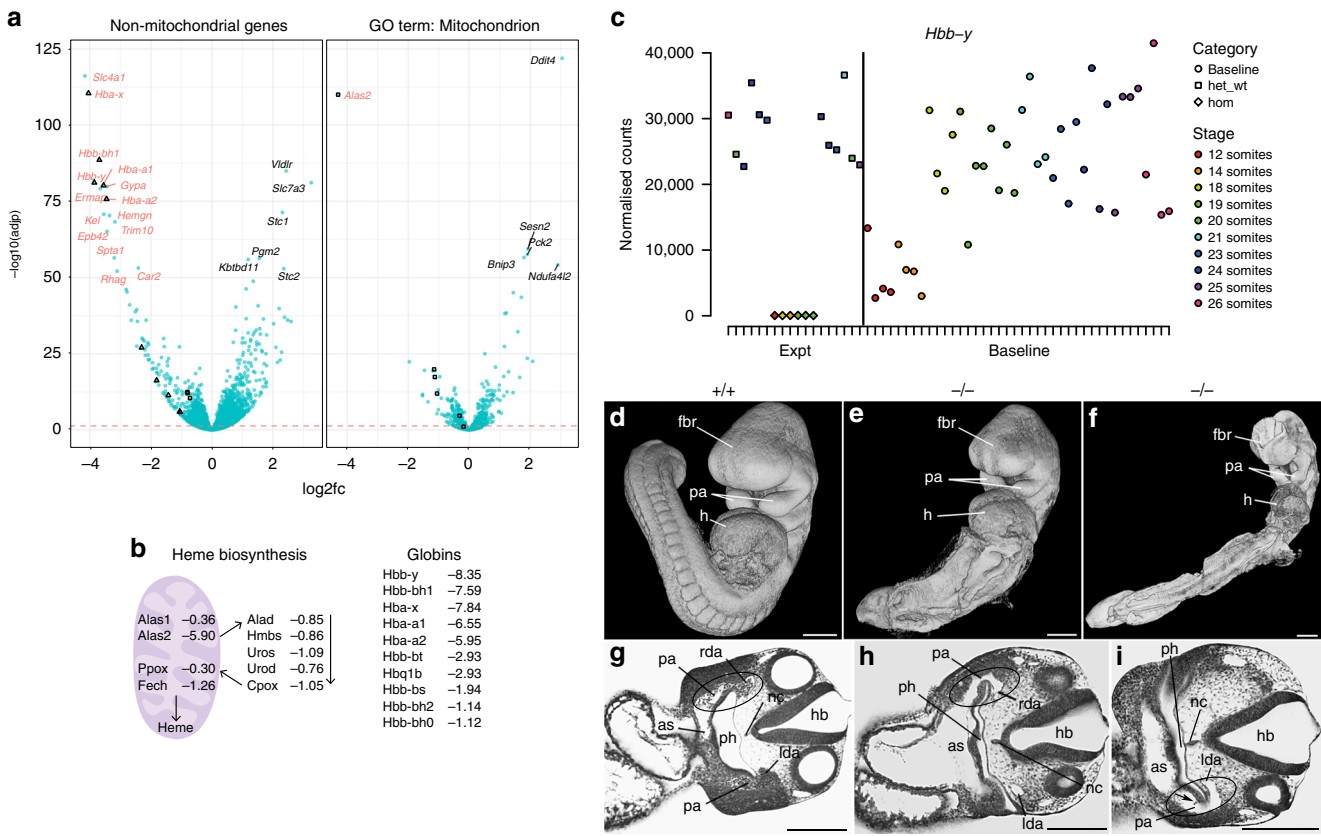

**Fig. 6** Phenotype of the *Nadk2* line. **a** Volcano plots of genes in the *Nadk2* mutant line. $\log_2$[fold change] is plotted on the *x*-axis and $-\log_{10}$[*p* value] on the *y*-axis. Genes associated with Gene Ontology term GO:0005739 (mitochondrion) are shown on the right and those not associated are on the left. Squares indicate genes from the heme biosynthesis pathway and triangles are haemoglobin genes. Gene names in red are associated with erythrocyte development. **b** Genes involved in the heme pathway on the left and haemoglobins on the right, plus their $\log_2$[fold change] in the Mutant Response. Genes whose products are localised to mitochondria are shown inside the purple oval. **c** Example count plot for *Hbb-y*, containing samples from the mutant line as well as somite stage-matched baseline samples. Homozygous embryos are shown as diamonds, siblings (heterozygous and wild-types) are squares and baseline samples are represented by circles. Points are coloured according to stage. **d–i** High-resolution episcopic microscopy data for the *Nadk2* line. **d–f** 3D models of the embryo surface. **d** Wild-type embryo (+/+). **e**, **f** Homozygous mutant embryos (−/−); **e** is mainly delayed in its caudal body parts, whereas **f** is globally delayed. fbr forebrain, h heart, pa pharyngeal arch. **g–i** Corresponding axial sections from the embryos in (**d–f**) at the level of the heart. as aortic sac, pa pharyngeal arch artery, lda left dorsal aorta, rda right dorsal aorta, ph pharynx, hb hindbrain (respectively fourth ventricle), nc notochord. The control embryo (**g**) has a large number of clearly visible blood cells in the aortic sac and arteries, whereas the homozygotes have no, **h** (4 of 6), or very few, **i** (2 of 6), erythrocytes (arrow). Compare circled areas in (**g**, **h** and **i**). Scale bars are 250 μm. Source data are provided as a Source Data file

causing delay in other tissues. To allow for this possibility we have included a No Delay category in our analysis. We find many genes that change expression over the baseline time course are not affected by delay, but are expressed at the same level in morphologically delayed embryos as in wild-type and heterozygous siblings. That is their expression is at the appropriate level for an E9.5 embryo rather than matching the stage as determined by their somite number. For example, in the *Fcho2* line, this is the case for genes involved in cardiovascular development such as *S1pr1* and *Pecam1* and nervous system genes such as *Gbx2* and *Wnt5b* (Supplementary Fig. 3a–b). This suggests that some tissues in certain mutants continue to develop at the same rate as in wild-type embryos despite the apparent global developmental delay as measured by somite number.

In our whole-organism RNA-seq analysis we detected the vast majority (97.0%) of genes identified in single-tissue RNA-seq at a comparable embryonic stage (Supplementary Fig. 1c), which demonstrates the sensitivity of whole-organism RNA-seq. In addition, DGE lists and their corresponding GO term enrichment for lines of known mutants establish this method as a valid screening tool. For example, human orthologues of five genes included in this study have a confirmed role in human

ciliopathies, often characterised by impaired Shh signalling and showed overlapping DGE lists. We identified two additional lines, *Kif3b* and *Kifap3*, that showed similar profiles and have morphological phenotypes such as polydactyly and *situs inversus*. Both proteins bind to Kif3a to form the kinesin 2 complex facilitating the anterograde transport of cargo proteins along cilia microtubules[63]. Taken together, this makes these genes strong candidates for idiopathic ciliopathies. Our analysis also implicates two non-coding transcripts and *Sox21* in Shh-related early developmental processes.

Finding members of the Shh pathway in all the DGE lists of ciliopathy genes in our collection makes us confident to predict possible functions for less well-described genes like *Zkscan17* (Fig. 5c) and *Hmgxb3* (Fig. 5d). Our expression profiles suggested the absence of erythrocytes in *Nadk2* mutants and this was confirmed by our morphological analysis. The absence of erythrocytes in *Nadk2* mutants could be a dissection artefact due to rupturing of large blood vessels; however, this is highly unlikely to occur in all six homozygous mutants, but not the wild-type siblings. Further work is necessary to ascertain whether the lack of red blood cells is causal for the global developmental delay in these mutants.

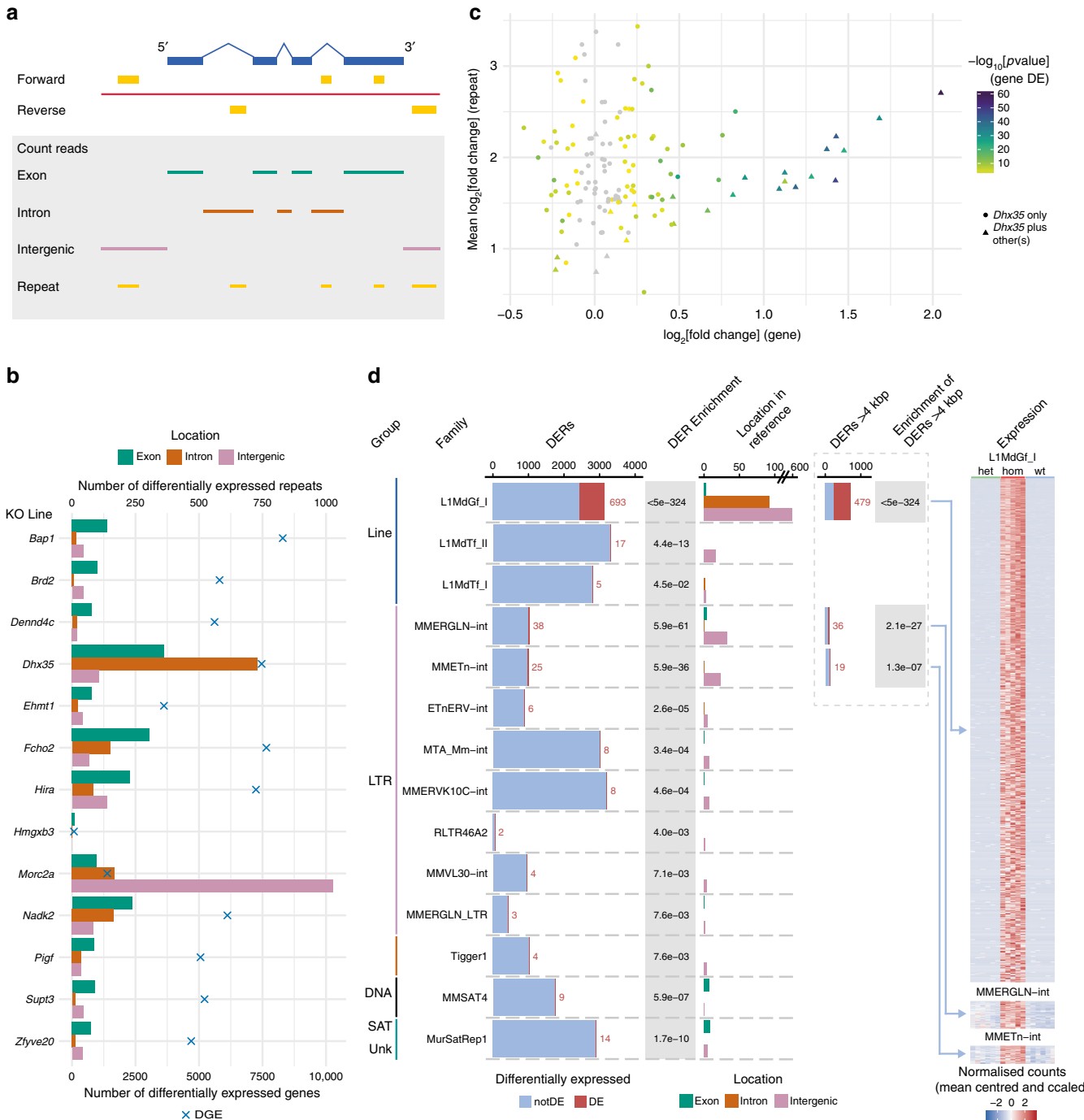

**Fig. 7** Investigation of repeat deregulation. **a** Schematic representing a region of the genome (red line), a gene on the forward strand (exons are blue boxes and introns are blue lines) and strand-specific repeats (yellow boxes). The area within the grey box shows the genomic stretch where strand-specific reads were considered within exons (green), within introns (brown), intergenic (purple) or within annotated repeats (yellow). **b** Differential repeat and gene expression for 13 mutant lines (*Hmgxb3* acts as a negative control). For each mutant line the number of differentially expressed repeat instances (DERs) is shown on the top *x*-axis split by genomic location (bars). The number of genes differentially expressed (homozygotes versus siblings) for each mutant line are indicated by a cross and shown on the bottom *x*-axis. *Dhx35* has a large number of DERs in intronic regions (brown bar), whereas for *Morc2a* they are mostly in intergenic regions (purple bar). **c** Fold change of repeats and genes in *Dhx35* mutant line. Genes enriched for the number of intronic DERs, if greater than 1 instance, compared to all introns for each gene are shown by circles (unique to *Dhx35*) or triangles (found in *Dhx35* and at least one other mutant), ENSMUSG00000107482 has no exon counts and was excluded from the plot. The colour shows the adjusted *p* value (−log$_{10}$ scaled) from DESeq2 for the gene with grey as not significant. The log$_2$[fold change] for the gene is on the *x*-axis with the mean fold change of intronic DERs on the *y*-axis. **d** Repeat instance expression in the *Morc2a* mutant line. Only repeat families enriched for individual DERs against total instances in the genome are shown. Bar shows total repeat instances with DERs in red. Bars in the Location in Reference column display the number of DERs by genomic location relative to gene annotation. Dashed box shows subset of repeat instances ≥4 kb. Mean centred and scaled heatmaps in the Expression column are of expression levels of DERs (L1MdGf_I, MMERGLN-int and MMETn-int families) across heterozygous (het), homozygous (hom) and wild-type (wt) *Morc2a* embryos. Source data are provided as a Source Data file

We found changes in large numbers of repeat elements in two of the mutant lines. Homozygous mutant *Dhx35* embryos displayed differentially abundant intronic repeat elements, the majority of which are SINEs, in the sense orientation. The most obvious explanation for this is that the genes containing the introns affected are also differentially expressed and the repeats appear affected as a consequence of sequencing unprocessed mRNA. This is unlikely since the majority of genes containing the DE repeat instances have much smaller expression changes which are often in the opposite direction. Our data suggest that specific introns are being retained at a higher rate in homozygous mutants than in wild-type embryos. Future in-depth analysis of this putative RNA helicase would be needed to better understand the function and mechanisms of intron retention.

In contrast to *Dhx35*, de-repressed repeat elements in homozygous *Morc2a* embryos were mostly in intergenic regions. L1s were most affected with over 20% of 3132 instances of L1MdGf_I family repeats de-repressed. This class of repeats was almost completely silent in wild-type embryos at E9.5. Notably, substantial numbers of DE L1MdGf_I repeats mapped to Chr Y, but were detected in female embryos. Repeat sequences are notoriously difficult to map and it is therefore possible this is caused by mis-mapping of reads or accumulated polymorphisms. Alternatively, this suggests the possibility that if, as has been suggested[60], these elements are capable of retrotransposition, they have been mobilised in the *Morc2a* KO background C57BL/6NTac/USA and instances now reside on autosomes, but are not present in the reference sequence of C57BL/6J. Ultimately only whole-genome sequencing of the *Morc2a* background will be able to resolve this. The LINE families L1MdGf and L1MdT and the LTR family MMERGLN-int lose their hypermethylation in sperm during the sperm to zygote transition which later gets reinstated by E6.5/7.5 [64]. Because we observed expression of these families at E9.5, our data suggest that *Morc2a* is involved in this developmental process. Similarly, a *Morc1*-deficient mouse has de-repressed transposable elements in the male germline including instances of the L1MdGf, MMERGLN and MMETn-int families[65].

To allow other researchers to view the data for the mutant lines presented here and apply our analysis method to their own data, we have produced a tool called Baseline CompaRe. This is designed to take count data and sample information files and run DESeq2 as in Supplementary Fig. 2. The results are presented as tables split into the four categories (Mutant Response, Delay, No Delay and Discard). Baseline CompaRe produces PCA plots and graphs of normalised counts for individual genes. It is available at https://www.sanger.ac.uk/science/tools/dmdd/dmdd/ or to download for use locally at https://github.com/richysix/baseline_compare. Also, we have provided a list of the genes that contribute most to the somite number signal as captured by PC3. These genes change the most over the time course and are therefore most prone to inappropriately appearing differentially expressed because the homozygous embryos are developmentally delayed (Supplementary Data 5).

This work creates a large transcriptomic resource, which together with the morphological phenotype information on these lines, will provide valuable information to researchers and further the understanding of the role of these genes in early mammalian development.

## Methods

**Animal care.** The care and use of all mice in this study were in accordance with UK Home Office regulations, UK Animals (Scientific Procedures) Act of 1986 (PPL 80/2485 and P77453634) and were approved by the Wellcome Sanger Institute's Animal Welfare and Ethical Review Body.

**Sample collection.** All mice were produced and maintained on a C57BL/6N genetic background. All embryos were produced by the Wellcome Sanger Institute (https://www.sanger.ac.uk/mouseportal/) as part of the DMDD project[11]. Gene knock-out lines were produced as part of a systematic programme coordinated by the International Mouse Phenotyping Consortium[16] and are available from http://www.mousephenotype.org. Lines were designated lethal if no homozygous mutants were present amongst a minimum of 28 pups at P14. For each line, one or more litters from heterozygous intercrosses were harvested at embryonic day E9.5. After culling the dam by cervical dislocation, embryos were dissected out of the uterus in cold PBS buffer, removed from the decidua and cleared of all surrounding membranes. Each embryo was genotyped using yolk sac samples, staged by counting the number of somites and placed into RNAlater® stabilisation solution overnight at 4 °C, before storing at −20 °C. Samples are detailed in Supplementary Data 1.

**Sample genotyping and sex determination.** The genotype of each sample—excluding those made by CRISPR or where the lacZ reporter gene was removed—was checked by aligning unmapped reads to lacZ (sequence extracted from https://www.i-dcc.org/imits/targ_rep/alleles/30096/escell-clone-cre-genbank-file) using BWA[66], counting the aligned reads and normalising with the same DESeq2 size factors as used for creating BioLayout input. Normalised counts were used to check which samples were wild type (lacking lacZ counts), heterozygous (intermediate lacZ counts) and homozygous (highest lacZ counts). The genotypes of 30/1098 (2.7%) samples were changed by this analysis and noted in the metadata submitted to the European Nucleotide Archive.

The sex of each sample was checked by assessing the normalised counts for *Xist* (ENSMUSG00000086503) and four Y chromosome genes with large expression level ranges (ENSMUSG00000069049, ENSMUSG00000068457, ENSMUSG00000069045, ENSMUSG00000056673). Female samples were defined as those with *Xist* counts above 400 and levels of ENSMUSG00000069049, ENSMUSG00000068457, ENSMUSG00000069045 and ENSMUSG00000056673 below 20, 6, 10 and 60 respectively. The sexes of 43 samples were changed by this analysis and noted in the metadata submitted to ENA.

**RNA extraction and sequencing.** Individual embryos were lysed in 300 µl Trizol with a 5 mm stainless steel bead (Qiagen) for 2 min at 20 Hz in a tissue lyser (Qiagen). After mixing of 200 µl chloroform to the homogenate and incubation for 30 min at room temperature the upper phase was transferred to a new tube. One volume of fresh 70% ethanol was added and the RNA was purified over a spin column (Qiagen RNeasy MinElute). After quantification (Qubit RNA BR) the sample was treated with DNAse enzyme (Qiagen) and purified over a spin column again. Up to 500 ng total RNA was used for the generation of strand-specific RNA-seq libraries containing unique index sequences in the adapter. Libraries were pooled and sequenced on Illumina HiSeq.

**RNA-seq analysis.** FASTQ files were aligned to the GRCm38 reference genome using TopHat2 [67] (v2.0.13, options: --library-type fr-firststrand). Ensembl 88 gene models were supplied to TopHat2 to aid transcriptome mapping. Sequence quality of each library was manually assessed one knock-out line at a time using the output of QoRTs[68]. Libraries were quality controlled on insert size, GC content and proper pairs. Counts for genes were produced using htseq-count[69] (v0.6.0 options: −stranded = reverse) with the Ensembl v88 annotation as a reference. For the comparison to RNA-seq on dissected tissues (PRJEB4513), data were downloaded from the Sequence Read Archive (SRA, https://www.ncbi.nlm.nih.gov/sra) and processed in the same way.

**Correlation networks.** Normalised counts obtained from DESeq2 were used to create a file for inputting into BioLayout *Express*3D [21,22]. The Pearson correlation cut-off was set to 0.8; gene pairs with a correlation coefficient above this are linked by an edge in the input graph (10,844 genes). After running MCL, clusters with fewer than five genes were removed (36 genes) which produced 111 clusters ranging in size from 6 to 2706 genes. Genes in the batch outlier cluster (Cluster006) were then removed in subsequent analyses. The networks in Supplementary Fig. 1 were produced by using the samples as nodes and setting the correlation coefficient cut-off to the maximum value that still includes all samples (0.96 and 0.97 for Supplementary Fig. 1a and b respectively).

**Incorporation of RNA-seq data into Ensembl.** RNA-seq models were also generated using our Ensembl RNA-seq pipeline[70]. Models were generated by assigning the samples to six time points (E8_dpc, E8.5_dpc, E9_dpc, E9.5_dpc, E10_dpc, E10.5_dpc). These data were aligned to the genome using BWA[66], resulting in 3,470,792,067 reads aligning from 3,726,475,010 reads. The aligned reads were processed by collapsing the transcribed regions into a set of potential exons. Partially aligned reads were re-mapped using Exonerate[71], which identified spliced reads and introns. These introns together with the set of transcribed exons were combined to produce transcript models, one set for each time point and one set produced by merging data from all of the time points. The longest open reading frame in each of these models was compared to UniProt protein sequences— UniProt protein existence levels 1 (existence at protein level) and 2 (existence at transcript level)—using BLAST[72] in order to classify the models according to their

protein-coding potential. The RNA-seq transcript models, indexed BAM files, and the complete set of splice junctions identified by our pipeline are available in Ensembl. The tracks can be switched on using the control panel (https://www.ensembl.org/info/website/control_panel.html).

**Novel genes assembly**. Cufflinks[73] and Cuffmerge were run on all samples and novel transcripts were identified as those transcripts that did not overlap existing annotation and that had multiple exons and canonical splice sites. To assess the coding potential of the novel genes, models in the same region were clustered to reduce redundancy. Transcripts that did not overlap current genes were scanned for pfam domains using InterProScan[74] (v5.25–64.0, https://www.ebi.ac.uk/interpro/search/sequence-search). Some of the genes have subsequently been annotated in later versions of Ensembl.

**Principal component analysis**. PCA used data on all 986 experimental samples including all genes after removal of the blacklist genes. The data were transformed using the variance stabilising transformation function provided by DESeq2[75] and the PCA was calculated using the prcomp function in R. The genes contributing most to PC3 (3872 genes that cumulatively contribute 50% of the variance captured by PC3) are provided in Supplementary Data 5.

**Differential expression**. Differential expression analysis was done using DESeq2[75] with the counts from htseq-count. Genes with an adjusted $p$ value (Wald test, Benjamini−Hochberg adjustment for multiple testing) below 0.05 were called as differentially expressed. For standard analysis (i.e. no delayed embryos), the model used was ~ sex+condition, which takes the sex of the embryos into account. For lines with delayed embryos DESeq2 was run in three ways:

(1) Experimental samples (hom, het and wt) with model as ~ sex + condition.
(2) Experimental samples and baseline samples of the stages present in the experimental embryos using the same model (~ sex + condition).
(3) Experimental samples and baseline samples of the stages present in the experimental embryos using a model that takes stage into account (~ sex + stage + condition).

The differentially expressed gene lists were then overlapped. Genes were placed into one of four categories based on which lists they appeared in (Fig. 2).

**Repeats**. Repeat annotation identified by RepeatMasker was downloaded (http://www.repeatmasker.org/genomes/mm10/RepeatMasker-rm405-db20140131/mm10.fa.out.gz) and filtered to remove repeats marked as Simple_repeat or Low_complexity. In order to reduce potential mapping bias, FASTQ files were realigned to the GRCm38 reference genome using TopHat2[67] (v2.0.13, options: --library-type fr-firststrand), but without a transcriptome reference. Counts for repeats were produced using htseq-count[69] (v0.6.0 options: --stranded = reverse). Intronic retention was examined using iDiffIR (https://bitbucket.org/comp_bio/idiffir) and custom scripts (https://github.com/iansealy/bio-misc/blob/master/analyse_intronic_expression.pl). The results files can be downloaded from Figshare (https://doi.org/10.6084/m9.figshare.7613528). The dataset contains a results file for each of the *Hmgxb3*, *Dhx35* and *Morc2a* lines.

**Anatomical enrichment**. EMAPA enrichment was done with Ontologizer[76] (http://ontologizer.de) using the Parent−Child−Union calculation method and Benjamini−Hochberg correction for multiple testing. The EMAPA ontology was downloaded from ftp://ftp.hgu.mrc.ac.uk/pub/MouseAtlas/Anatomy/EMAPA.obo and the annotations of EMAPA ids to MGI gene ids was downloaded from MouseMine[77] (www.mousemine.org/) using a custom python script (available from the code repository—see Code Availability). These data were filtered to use only data from wild-type samples with the following strength annotations (Present, Moderate, Strong, Weak, Very strong) and assay types (RNA in situ). MGI gene ids were converted to Ensembl gene ids by downloading the mapping of one to another from Ensembl BioMart (https://www.ensembl.org/biomart) and removing duplicate mappings. To simplify the enriched terms, they were aggregated using the EMAPA ontology graph. A set of terms close to the top of the ontology graph were chosen as parent terms, such as hindbrain, branchial arch and skeletal system. For each parent term, all its child terms were aggregated together to produce the bubble plots.

**GO enrichment**. GO enrichment was performed using the R topGO package[78]. The mapping between Ensembl gene ids and GO terms was retrieved from the Ensembl database using a custom Perl script (get_ensembl_go_terms.pl) from the topgo-wrapper repository (https://github.com/iansealy/topgo-wrapper).

**Enrichment Map**. The Cytoscape[79] Enrichment Map plugin[80] was used to visualise the GO enrichment results (http://www.baderlab.org/Software/EnrichmentMap).

**Reporting summary**. Further information on research design is available in the Nature Research Reporting Summary linked to this article.

## Data availability

The authors declare that all data supporting the findings of this study are available within the article and its supplementary information files or from the corresponding author upon reasonable request. All sequencing data have been deposited in ENA under accession codes: ERP010754 (baseline) and ERP008773 (mutant lines). Processed data such as DGE lists and GO/EMAPA enrichments for each mutant line are available at Figshare (https://doi.org/10.6084/m9.figshare.c.4127441). The source data underlying Figs. 1d, 2a, c−e, 3b−d, 4a−c, 5c, 6b−d, and Supplementary Figs. 1d, 2, 3, 4a, b, 5a, b, d−f, 6 are provided as a Source Data File.

## Code availability

All the codes used to produce the plots are available at https://github.com/richysix/dmdd_figs. The Baseline CompaRe code is available at https://github.com/richysix/baseline_compare.

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

## Acknowledgements

We are grateful to Christopher Dooley, Terry Meehan and Kate Carpenter for critical reading of the manuscript and to Chris Lelliott and Thomas Keane for helpful discussions. We would like to thank the Wellcome Sanger Institute sequencing pipelines for performing sequencing and the staff of the Research Support Facility for mouse care, as well as all contributors to the DMDD programme. This work was supported by Wellcome Trust Strategic Award WT100160MA, by the Wellcome Sanger Institute (WT098051,206194), and the European Molecular Biology Laboratory.

## Author contributions

J.E.C., R.J.W. and I.M.S. performed formal analysis of the data. N.S., I. Packham and N.W. performed experiments. C.T., C.M., A. Green, E.S., E. Ryder, J.K.W. and A. Galli performed all mouse colony management, breeding, sample collection and genotyping

work. I. Papatheodoru, A.T., A.F. and K.B. processed data for Expression Atlas and Ensembl integration. W.J.W. and S.H.G. did phenotype analyses. J.E.C., D.L.S., R.J.W., I. M.S. and E.M.B.-N. designed the study and analysis methods with conceptual advice from M.H., E. Robertson, J.C.S., T.M. and D.J.A. J.E.C., R.J.W., N.S. and E.M.B.-N. wrote the manuscript with advice from I.M.S., M.H., and D.J.A.

## Additional information

**Competing interests:** The authors declare no competing interests.

