## [Peer Review File · Nature Communications]

Reviewers' Comments:

Reviewer #1:

Remarks to the Author:

The manuscript presented by the DMDD consortium describes the whole embryo RNA-seq analysis of 73 mutant lines that show lethality at P14. The authors obtained the transcriptomes of homozygous embryos, if available, and of E9.5 heterozygous embryos and wild type littermates. Since the transcriptomes of whole embryos change rapidly during development, the authors established a "baseline" of gene expression of embryos at stage 4-28 and 34-36 somites. Therefore mutant embryos could be compared to wild-type embryos of the matching somite stage. 20 of 73 lines revealed no homozygotes at E9.5; these were early lethal and could only be analysed by comparing hets with wild type. 35 lines showed no DGE in hom vs sibs or het vs wt after filtering (Fig. 3C). 36 lines could be analysed on the basis of DGE. Among those a group of genes was identified, which have been linked to human ciliopathies previously. The transcriptome profiles of these mutants were similar. Two more mutants with similar profiles have been linked to Shh signaling. Thus, by linking previous knowledge with DGE profile overlaps a larger group of genes acting in developmental processes related to Shh signaling could be identified. This does not reveal the role of each new member of the group in embryogenesis, but at least provides a strong hint what to look for.

Transcription analysis outside exons revealed derepression of repeats in a few mutants.

The manuscript reveals the strength and weakness of the approach. DGE analysis of homozygous mutant embryos with hets and exactly stage matched wild type embryos can provide important clues with respect to the possible cause of the mutant phenotype if embryos are analyzed at or close to the onset of the mutant phenotype. However, since lethality at P14 was chosen for selection of mutant lines and the transcriptome analysis was performed at E9.5, it is not surprising that half of the lines either produced no homozygous embryos at E9.5 (early lethal) or showed no strong DGE after filtering (later lethal). The further the lethal stage is away from the stage examined, E9.5, the less informative is the transcriptome data.

The identification of a gene group related to Shh signaling, which was presented as best example for the power of this approach with respect to relating genes to function in embryogenesis was based on substantial prior knowledge of the phenotypes of several genes in that group. The possible function of about 30 more genes, for which DGE profiles could be determined, however, remains obscure.

Overall the consortium produced an enormous body of valuable information and evaluation tools which will be useful for many groups analyzing mutants generated on the same genetic background, in particular those showing lethality around E9.5. The work adds an important corner stone to a portfolio of complementing approaches aiming at analyzing the function of all genes of the mouse. It should be published.

Comments:

- Abstract, Fig 1d + text: "somitogenesis" is not the correct term here; it describes the process of somite formation from presomitic mesoderm. Here, somite stage during trunk development should be used for description and indicated by the somite number.

- Figure 3 should be explained better in the main text.

- Figure 4e, f related to Nadk2 mutant: the authors "suggest a substantial reduction or absence of erythrocytes in the homozygous embryos" (cit.). This is an excellent case for validation of the approach. Is the prediction based on molecular data actually guiding to the cause of lethality? If yes, it would strengthen the approach and the manuscript.

- The retrotransposon part appears disproportionately large
- The mouse inbred strain used must be mentioned in the text.
- Pitx2 appears twice on Figure 4b and with different profiles; what is the reason?

Reviewer #2:

Remarks to the Author:

This work from Collins et al analyzes the whole embryo transcriptomes of about 53 separate homozygous lethal mouse mutant lines at E9.5 with the goal of “uncover[ing] transcriptional events underlying embryonic lethality.” In addition, they work to identify a transcriptional signature of developmental delay (as assessed by retarded somitogenesis). This work is conceptually innovative and of potential great use to the mouse genetics community. In particular, a computational tool that allowed researchers to separate out the developmental delay transcriptional signature from more mutation-specific signatures, even at a single stage of mouse development, would be a boon. However, the tools to do this are not described in sufficient detail or provided through an open source for other researchers to be able to do so. A challenge, and promise, for computational biology is to make approaches sufficiently detailed and accessible for other researchers to replicate the findings from the raw data and build on the approach. Neither is possible in the case of this manuscript.

As for the second laudable goal of identifying transcriptional events causing lethality, a few tantalizing vignettes are described for a subset of mutants, but no biological validation is done of any of the hypotheses gleaned by transcriptome gazing. Thus, no causes of lethality can be described as being identified, nor do these vignettes rise beyond the level of initial hypothesis.

For these two reasons, I do not support publication. However, the promise of this work is great, and if it were to be considerably expanded to provide either a tool for analyzing E9.5 transcriptomes for evidence of delay and subtraction of the delay signature, or evidence that transcriptomic analysis of many mutants provided biological insights beyond what can be arrived at from individual transcriptomes, this work would be of substantial help to a large part of the computational and developmental biology communities.

Minor points:

The figures are hard to read, incorporate fonts at wildly disparate sizes, and are insufficiently described in their legends. As just one example, the authors state in the legend for Fig 2 that they collected “baseline samples of the appropriate stages,” but it is not clear what they deemed to be the appropriate stages or how these were defined. I assume this was based on somite-stage, but the reader should not be left to guess at what was done, nor how it was done. Other panels are minimally-informative hairballs, or appear to reflect insufficient scrutiny of computational outputs (e.g., why are there two lines for Tbx5 in Figure 4?).

The manuscript needs substantial attention to the precision of the text. For example, the authors write that “We collected embryos at E9.5 from 73 lines that were homozygous lethal by P14 (Fig. 1a).” However, they only attempted to collect from 73 lines at this stage and collected from fewer. Or the authors describe “synapse-located Slitrk genes.” To my knowledge, the genes are not located in the synapse. Or the authors state that, “The DE repeat instances reside in 472 genes (21 instances in more than one gene).” This statement appears to use “gene” to indicate two different things within the same sentence, sowing confusion.

Reviewer #3:

Remarks to the Author:

The manuscript by Collins and colleagues provides the latest update in the ongoing Deciphering the Mechanisms of Developmental Disorders (DMMD) program, a project aimed at mass phenotyping of hundreds of mutant mouse lines. In this study, the group analyzes whole embryo transcriptomes collected from more than 100 embryos, representing more than 50 different gene mutations, during somitogenesis (E8.0-9.5). The study provides a valuable resource for the community, as the raw data will be added to Ensembl. Additionally, the analysis provides several interesting new insights. First, the analysis reveals a common transcriptional signature, that the authors surmise is due to developmental delay, a more generic phenotype observed among disparate mutant embryos, which is potentially valuable to other researchers. Second, the analysis reveals terms and pathways associated with each individual mutant. Finally, the analysis reveals novel transcripts that do not correspond to coding region, which is potentially interesting to some, although lacking in mechanism.

The study represents an ambitious effort and potentially valuable resource. Several comments and questions below may help the authors produce a more readable and rigorous study suitable for publication in Nature Communications.

Major Points:

1. The authors use somite number to stage all embryos, including mutants. However, some of the mutants could exhibit a phenotype that influences somite number without alternating developmental progression. This could impose a bias in the transcriptional analysis. The authors should comment on whether this alternative interpretation was considered.
2. The authors do not provide detail regarding the PCA that reveals a developmental delay. Were all genes included? Or just the most differentially expressed? In addition, the list of genes comprising the developmental delay should be provided.
3. The final section of the Results (line 211 and on) is difficult to follow and seems topically unrelated to the rest of the manuscript.

Minor Points:

1. The Introduction is not as compelling as it could be. In places, it is hard to know if the authors are describing work done in this study or previously (e.g., line 58).
2. Line 79 – results heading. “during somitogenesis” would be more accurate than “of somitogenesis” since the analysis does not exclusively focus on the process of somitogenesis. Rather, the analysis includes all tissues and cells and their transcriptional changes during the process of somitogenesis.
3. Line 96 “novel mouse genes” – does this mean genes that are unique to mice and not found in other mammalian species? Or unannotated mouse genes?
4. Fig. S1 – it would be helpful if the nodes were labeled or annotated.
5. Lines 169-171 – where is this analysis shown?
6. Line 213 and on, why is the term “repeat” used? Do you mean expressed sequence? Or repetitive genome element?
7. Line 238-239 – what is meant by “significant?” If statistical, please provide evidence.

Reviewer #4:

Remarks to the Author:

Summary

The manuscript “Common and distinct transcriptional signatures of mammalian embryonic lethality” by John et al have characterized the transcriptome of 73 mouse mutant lines, which were homozygous lethal by P14, at E9.5 stage at the whole-embryo level. The authors firstly produced a comprehensive baseline dataset of wild-type embryos. Then, by exploring DGE in three ways, the authors could distinguish delay, no-delay, and mutant expression signatures. Using this approach, the authors identified the mutant response genes and performed deeper analysis of these genes for some mutant lines relating to SHH signal pathway, brain development, cell cycle heme

biosynthesis and repeat deregulation. These data provide a valuable resource for understanding the functions of mammalian embryonic essential genes. .

The authors performed a 3-ways differential gene expression (DGE) analysis, based on their comprehensive baseline dataset of wild-type embryos. In general, I think that the approach is reasonable to separate embryonic delay signatures from mutant line-specific transcriptional changes. They have also shown that the approach can indeed make the signal of the specific mutant response stronger. However, I have some questions about how they analyzed the data. I also think that more experiments are need to validate their data. I arranged my questions in figures.

Major:

1. Figure 1. The authors remove 194 genes as technical batch signals. What kinds of gene are them? Are they represent biological or technical batch signals? Also, how to make sure the vast majority of batch effects have been remove? Since this analysis is based on the data of the normal embryos, how to remove the batch effect of the mutant embryos? Why just removing these genes but not use software like SVA to remove the batch effect?
2. It is reasonable to remove genes in Y chromosome and Xist to avoid confounding of sex. However, why to move the mitochondrial genes, which may be involved in mutation mechanisms.
3. Figure 2. In Fig2b, why the genes overlapped between B and C, but not A, were also considered as mutant response genes, regarding that they are not differentially expressed between the homozygous mutants and their siblings?
4. In line 130-131, the authors said that the "No delay" genes is the same as in siblings, but in Fig2b, the "No delay" genes include a part of genes within A (which are differentially expressed genes between homozygous mutants and silblings), why?
5. Would the authors draw a diagrammatic sketch to show more clearly the three ways they run the DGE analysis?
6. Are the "Delay" genes similar among mutant lines? I suggest the authors draw a heatmap to show all the differentially expressed genes.
7. Figure 4-5. All the analysis of this manuscript is based on the RNA-Seq data. However, independent experiments including immunostaining should be performed to verify some results. For example, the results in Figure 4b suggested that Sox21 is new target gene of SHH pathway. This results should be verified regarding that the phenotype of Sox21 mutant seems not associated with SHH pathway (PMID: 19470461).

Minor:

1. What does "expt" mean in fig2c?
2. Figure 2c. It is better to make clear in the legend that the samples within the circles are homozygous mutants.

We would like to thank the reviewers for taking the time to review our manuscript and the helpful comments and suggestions.

Reviewer #1 (Remarks to the Author):

The manuscript presented by the DMDD consortium describes the whole embryo RNA-seq analysis of 73 mutant lines that show lethality at P14. The authors obtained the transcriptomes of homozygous embryos, if available, and of E9.5 heterozygous embryos and wild type littermates. Since the transcriptomes of whole embryos change rapidly during development, the authors established a “baseline” of gene expression of embryos at stage 4-28 and 34-36 somites. Therefore mutant embryos could be compared to wild-type embryos of the matching somite stage. 20 of 73 lines revealed no homozygotes at E9.5; these were early lethal and could only be analysed by comparing hets with wild type. 35 lines showed no DGE in hom vs sibs or het vs wt after filtering (Fig. 3C). 36 lines could be analysed on the basis of DGE. Among those a group of genes was identified, which have been linked to human ciliopathies previously. The transcriptome profiles of these mutants were similar. Two more mutants with similar profiles have been linked to Shh signaling. Thus, by linking previous knowledge with DGE profile overlaps a larger group of genes acting in developmental processes related to Shh signaling could be identified. This does not reveal the role of each new member of the group in embryogenesis, but at least provides a strong hint what to look for.

Transcription analysis outside exons revealed derepression of repeats in a few mutants.

The manuscript reveals the strength and weakness of the approach. DGE analysis of homozygous mutant embryos with hets and exactly stage matched wild type embryos can provide important clues with respect to the possible cause of the mutant phenotype if embryos are analyzed at or close to the onset of the mutant phenotype. However, since lethality at P14 was chosen for selection of mutant lines and the transcriptome analysis was performed at E9.5, it is not surprising that half of the lines either produced no homozygous embryos at E9.5 (early lethal) or showed no strong DGE after filtering (later lethal). The further the lethal stage is away from the stage examined, E9.5, the less informative is the transcriptome data.

The identification of a gene group related to Shh signaling, which was presented as best example for the power of this approach with respect to relating genes to function in embryogenesis was based on substantial prior knowledge of the phenotypes of several genes in that group. The possible function of about 30 more genes, for which DGE profiles could be determined, however, remains obscure.

Overall the consortium produced an enormous body of valuable information and evaluation tools which will be useful for many groups analyzing mutants generated on the same genetic background, in particular those showing lethality around E9.5. The work adds an important corner stone to a portfolio of complementing approaches aiming at analyzing the function of all genes of the mouse. It should be published.

Response: We would like to thank the reviewer for the strong endorsement of our work.

Comments:

- Abstract, Fig 1d + text: “somitogenesis” is not the correct term here; it describes the process of somite formation from presomitic mesoderm. Here, somite stage during trunk development should be used for description and indicated by the somite number.

Response: We'd like to thank the reviewer for pointing this out. We have changed wording and figure accordingly.

- Figure 3 should be explained better in the main text.

Response: We have extended the description of Figure 3 in the main text.

- Figure 4e, f related to Nadk2 mutant: the authors “suggest a substantial reduction or absence of erythrocytes in the homozygous embryos” (cit.). This is an excellent case for validation of the approach. Is the prediction based on molecular data actually guiding to the cause of lethality? If yes, it would strengthen the approach and the manuscript.

Response: Following the reviewer’s suggestion, we have analysed the HREM (High Resolution Episcopic Microscopy) data available for this mutant (<https://dmdd.org.uk/mutants/Nadk2>). As predicted by the transcriptional profile, erythrocytes could not be detected in the large blood vessels of 4 of the 6 homozygous embryos while the remaining two showed only scattered erythrocytes. Wild-type littermates show normal amounts of erythrocytes. We have added these findings to the manuscript with images in Figure 4.

- The retrotransposon part appears disproportionately large

Response: In light of increasing evidence of the importance of repeat elements and their control in processes such as gene regulation, germ line differentiation and cancer (Elbarbary et al. 2016, Science - PMID:26912865; Barau et al. 2016, Science - PMID:27856912; Burns, 2017 Nat Rev Cancer - PMID:28642606) we feel this is an important part of the manuscript. However, we have thoroughly reworked this section to make it easier to read and more concise.

- The mouse inbred strain used must be mentioned in the text.

Response: We apologise for this omission and have added the strain to the Methods section.

- Pitx2 appears twice on Figure 4b and with different profiles; what is the reason?

Response: This was a manual copy and paste error during incorporation of the heatmap into the figure. We apologise for this error and have fixed it.

Reviewer #2 (Remarks to the Author):

This work from Collins et al analyzes the whole embryo transcriptomes of about 53 separate homozygous lethal mouse mutant lines at E9.5 with the goal of “uncover[ing] transcriptional events underlying embryonic lethality.” In addition, they work to identify a transcriptional signature of developmental delay (as assessed by retarded somitogenesis). This work is conceptually innovative and of potential great use to the mouse genetics community. In particular, a computational tool that allowed researchers to separate out the developmental delay transcriptional signature from more mutation-specific signatures, even at a single stage of mouse

development, would be a boon. However, the tools to do this are not described in sufficient detail or provided through an open source for other researchers to be able to do so. A challenge, and promise, for computational biology is to make approaches sufficiently detailed and accessible for other researchers to replicate the findings from the raw data and build on the approach. Neither is possible in the case of this manuscript.

Response: We're glad to see that the reviewer appreciates the novelty and value of this dataset. We would like to thank the reviewer for the suggestion to turn our analysis into a tool that is usable by the community without reprocessing our baseline. As described in more detail further below we have developed a Shiny app for this purpose.

As for the second laudable goal of identifying transcriptional events causing lethality, a few tantalizing vignettes are described for a subset of mutants, but no biological validation is done of any of the hypotheses gleaned by transcriptome gazing. Thus, no causes of lethality can be described as being identified, nor do these vignettes rise beyond the level of initial hypothesis.

Response: This work analyses the transcriptional profiles of over 70 different mutant lines for functionally meaningful enrichments using bioinformatics approaches. The size of the dataset means that we can only highlight a subset of the findings. However, we provide all DGE lists, GO and anatomical enrichments, and count data for further analysis and discovery by the research community. The Shiny app, together with the display of the data in Expression Atlas, allows access to the data also for non-bioinformaticians. Furthermore, we have validated the transcriptional results of the Nadk2 line using HREM data, which is now included in Figure 4.

For these two reasons, I do not support publication. However, the promise of this work is great, and if it were to be considerably expanded to provide either a tool for analyzing E9.5 transcriptomes for evidence of delay and subtraction of the delay signature, or evidence that transcriptomic analysis of many mutants provided biological insights beyond what can be arrived at from individual transcriptomes, this work would be of substantial help to a large part of the computational and developmental biology communities.

Response: We have developed a Shiny app to allow our baseline data to be used in conjunction with other E9.5 transcriptomic data. It also enables visualisation of the data presented in the manuscript. The app takes count and sample information for the experimental samples and analyses it along with the baseline data. DESeq2 is run in the three ways described in the paper and produces PCA plots, results tables and count plots, all of which can be downloaded. Since including somite stage information, and using single embryos, is critical for this approach and might not be available for some experiments, we have also produced a list of the genes that contribute most to the delay signal, which is provided as Supplementary Table 5. These genes are the most likely to appear differentially expressed as a result of developmental delay. This list can be used to flag genes to be wary of in datasets that do not have somite stage information collected.

Minor points:

The figures are hard to read, incorporate fonts at wildly disparate sizes, and are insufficiently described in their legends.

Response: We have edited figures to make font sizes as consistent as possible while maintaining readability. We have made figure legends more extensive to better describe the figures.

As just one example, the authors state in the legend for Fig 2 that they collected “baseline samples of the appropriate stages,” but it is not clear what they deemed to be the appropriate stages or how these were defined. I assume this was based on somite-stage, but the reader should not be left to guess at what was done, nor how it was done.

Response: We have corrected this to say stage-matched baseline samples.

Other panels are minimally-informative hairballs, or appear to reflect insufficient scrutiny of computational outputs (e.g., why are there two lines for Tbx5 in Figure 4?).

Response: The networks in Supplementary Figure 1 serve to show that samples cluster together after gene outlier removal. We have added arrows to the figure and text to the figure legend to make this clearer. We have corrected the copy-paste error of duplicating Pitx2 in Fig. 4 b.

The manuscript needs substantial attention to the precision of the text. For example, the authors write that “We collected embryos at E9.5 from 73 lines that were homozygous lethal by P14 (Fig. 1a).” However, they only attempted to collect from 73 lines at this stage and collected from fewer.

Response: We did collect embryos from 73 lines, but genotyping after the collection showed that 20 lines only produced heterozygous or wild-type embryos. We have clarified this in the text.

Or the authors describe “synapse-located Slitrk genes.” To my knowledge, the genes are not located in the synapse.

Response: We have corrected this to “transcripts of three genes coding for synapse-located Slitrk proteins”.

Or the authors state that, “The DE repeat instances reside in 472 genes (21 instances in more than one gene).” This statement appears to use “gene” to indicate two different things within the same sentence, sowing confusion.

Response: Repeat instances can reside in two genes if the genes are overlapping on opposite strands, which was meant here. However, we have removed this phrasing to avoid any confusion between genes and repeats.

Reviewer #3 (Remarks to the Author):

The manuscript by Collins and colleagues provides the latest update in the ongoing Deciphering the Mechanisms of Developmental Disorders (DMMD) program, a project aimed at mass phenotyping of hundreds of mutant mouse lines. In this study, the group analyzes whole embryo transcriptomes collected from more than 100 embryos, representing more than 50 different gene mutations, during somitogenesis (E8.0-9.5). The study provides a valuable resource for the community, as the raw data will be added to Ensembl. Additionally, the analysis provides several

interesting new insights. First, the analysis reveals a common transcriptional signature, that the authors surmise is due to developmental delay, a more generic phenotype observed among disparate mutant embryos, which is potentially valuable to other researchers. Second, the analysis reveals terms and pathways associated with each individual mutant. Finally, the analysis reveals novel transcripts that do not correspond to coding region, which is potentially interesting to some, although lacking in mechanism.

The study represents an ambitious effort and potentially valuable resource. Several comments and questions below may help the authors produce a more readable and rigorous study suitable for publication in Nature Communications.

We would like to thank the reviewer for the positive assessment of our work.

Major Points:

1. The authors use somite number to stage all embryos, including mutants. However, some of the mutants could exhibit a phenotype that influences somite number without altering developmental progression. This could impose a bias in the transcriptional analysis. The authors should comment on whether this alternative interpretation was considered.

Response: We have taken this possibility into account. It is for this reason that we have a separate category in our analysis called “No delay”. This category identifies genes that are not affected by the apparent delay, i.e. their expression is at the appropriate level for an E9.5 embryo rather than matching the apparent stage as determined by their somite number. Furthermore, we would expect that a phenotype exclusive to somites would be reflected in the GO and anatomical enrichment analysis.

2. The authors do not provide detail regarding the PCA that reveals a developmental delay. Were all genes included? Or just the most differentially expressed? In addition, the list of genes comprising the developmental delay should be provided.

Response: All genes and all experimental samples (all genotypes) were included. The counts were transformed using the variance stabilising transform function provided by DESeq2 and the PCA was calculated using the R prcomp function. We have updated the methods to include this information and have included a Supplementary Table of the genes which contribute most to PC3. It includes 3872 genes that cumulatively contribute 50% of the variance captured by PC3.

3. The final section of the Results (line 211 and on) is difficult to follow and seems topically unrelated to the rest of the manuscript.

Response: Given the importance of repeat elements in gene regulation and disease, we feel this is a crucial analysis. However, we agree that this section is very complex. We have shortened and rephrased these paragraphs to make them easier to follow.

Minor Points:

1. The Introduction is not as compelling as it could be. In places, it is hard to know if the authors are describing work done in this study or previously (e.g., line 58).

Response: We have edited the introduction to make it clearer.

2. Line 79 – results heading. “during somitogenesis” would be more accurate than “of somitogenesis” since the analysis does not exclusively focus on the process of somitogenesis. Rather, the analysis includes all tissues and cells and their transcriptional changes during the process of somitogenesis.

Response: We’d like to thank the reviewer for pointing this out. We have removed reference to somitogenesis from the text and figures.

3. Line 96 “novel mouse genes” – does this mean genes that are unique to mice and not found in other mammalian species? Or unannotated mouse genes?

Response: This means unannotated mouse genes, we have rephrased that section.

4. Fig. S1 – it would be helpful if the nodes were labeled or annotated.

Response: We have labelled the nodes.

5. Lines 169-171 – where is this analysis shown?

Response: This is based on comparing the enrichment of GO terms from the mutant response lists. The plot in Fig. 4a is clustered by the overlap of the terms between each pair of mutant lines. Kifap3 clusters with the 5 genes linked to human ciliopathies (grey box). Since Kif3b is also a component of the kinesin 2 complex we included it in the analysis shown in Fig. 4b. We have altered the text to make this clearer.

6. Line 213 and on, why is the term “repeat” used? Do you mean expressed sequence? Or repetitive genome element?

Response: We mean repetitive genomic elements that are expressed. The repeats used are defined in the methods section and we have clarified in the text that these are repeats as defined by RepeatMasker.

7. Line 238-239 – what is meant by “significant?” If statistical, please provide evidence.

Response: We now provide the evidence. The results files can be downloaded from Figshare. We have added a link in the paper.

Reviewer #4 (Remarks to the Author):

Summary

The manuscript “Common and distinct transcriptional signatures of mammalian embryonic lethality” by John et al have characterized the transcriptome of 73 mouse mutant lines, which were homozygous lethal by P14, at E9.5 stage at the whole-embryo level. The authors firstly produced a comprehensive baseline dataset of wild-type embryos. Then, by exploring DGE in three ways, the authors could distinguish delay, no-delay, and mutant expression signatures. Using this approach, the authors identified the mutant response genes and performed deeper analysis of these genes for some mutant lines relating to SHH signal pathway, brain development, cell cycle heme biosynthesis and repeat deregulation. These data provide a valuable resource for understanding the functions of mammalian embryonic essential genes.

The authors performed a 3-ways differential gene expression (DGE) analysis, based on their comprehensive baseline dataset of wild-type embryos. In general, I think that the approach is reasonable to separate embryonic delay signatures from mutant line-specific transcriptional changes. They have also shown that the approach can indeed make the signal of the specific mutant response stronger. However, I have some questions about how they analyzed the data. I also think that more experiments are need to validate their data. I arranged my questions in figures.

Major:

1. Figure 1. The authors remove 194 genes as technical batch signals. What kinds of gene are them? Are they represent biological or technical batch signals? Also, how to make sure the vast majority of batch effects have been remove? Since this analysis is based on the data of the normal embryos, how to remove the batch effect of the mutant embryos? Why just removing these genes but not use software like SVA to remove the batch effect?

Response: A large number of the 194 genes are histone cluster genes (41) and predicted genes (81). Histone cluster genes are not polyadenylated and so should not be present in poly-A pulldown RNA-seq. Also, the samples that show this signal were all prepared together. For these reasons, we believe this to be a technical artefact. Once these genes have been removed, these samples cluster together with all of the other samples in the BioLayout correlation network (Supplementary Fig. 1a-b). We believe this shows that no large batch effects remain. Removing these genes is simpler than using SVA and allows others to use the data without needing to correct for this batch effect.

2. It is reasonable to remove genes in Y chromosome and Xist to avoid confounding of sex. However, why to move the mitochondrial genes, which may be involved in mutation mechanisms.

Response: The genes that we have removed are only those expressed from the mitochondrial genome (37 genes) and not those expressed from the nuclear genome. We have made this clearer in the text. These genes were removed because they tend to be highly variable (see plot below) across the baseline samples and are therefore prone to appearing differentially expressed by chance depending on the experimental embryos sampled.

Expression of mitochondrial genes in baseline samples. Normalised counts for mitochondrial genes were mean-centred and scaled per gene across all baseline samples. Plot displays the mean value in each sample. Error bars are standard deviation.

3. Figure 2. In Fig2b, why the genes overlapped between B and C, but not A, were also considered as mutant response genes, regarding that they are not differentially expressed between the homozygous mutants and their siblings?

Response: Adding in the baseline samples provides a better assessment of the dispersion for each gene and therefore additional genes are detected as differentially expressed. We think these genes represent true differentially expressed genes. These genes are false negatives when only the experimental samples are considered.

4. In line 130-131, the authors said that the “No delay” genes is the same as in siblings, but in Fig2b, the “No delay” genes include a part of genes within A (which are differentially expressed genes between homozygous mutants and siblings), why?

Response: The genes that are called as being differentially expressed in both A and C, but not B, tend to be genes that are expressed at similar levels to the sibling samples, but at different levels to the stage matched baseline samples. This fits our definition of “No delay” genes.

For example, the gene *Atpv61f* (ENSMUG00000004285, see plot below) is expressed at slightly lower levels than siblings in the homozygous embryos in the Brd2 line. Without considering the baseline embryos, this difference is significant. However, the baseline embryos at the matching stages have much higher expression and appear to be more variable (for example the 28 somite stage embryo with very high expression). Therefore, the difference is no longer significant when the baseline samples are added. When stage is included into the model the homozygous samples are sufficiently different from baseline that the difference again becomes significant.

The adjusted pvalues for the DESeq2 runs in A, B and C are 0.0138, 0.0858 and 0.0035 respectively. Overall the mean adjusted pvalues for this group in Brd2 for A, B and C are 0.0302, 0.0943 and 0.0215 demonstrating that these genes are generally very close to the significance threshold of 0.05. If the size of the difference between homozygotes and siblings were larger or the gene less variable in the baseline embryos, this gene could be classified as Mutant Response.

5. Would the authors draw a diagrammatic sketch to show more clearly the three ways they run the DGE analysis?

Response: We’d like to thank the reviewer for this suggestion. We have included a diagram as Supplementary Figure 2.

6. Are the “Delay” genes similar among mutant lines? I suggest the authors draw a heatmap to show all the differentially expressed genes.

Response: We show in Supplementary Figure 3 that there is strong overlap between delayed lines with respect to affected tissues. We also state in the text that, in contrast, there is little overlap in the Delay DGE lists. We have now included in Supplementary Figure 3 a plot showing

this. Across all lines there are 10992 genes that appear in at least one Delay list. In our opinion, a heatmap across 986 samples and 10992 genes would be too large to be informative.

7. Figure 4-5. All the analysis of this manuscript is based on the RNA-Seq data. However, independent experiments including immunostaining should be performed to verify some results. For example, the results in Figure 4b suggested that Sox21 is new target gene of SHH pathway. This results should be verified regarding that the phenotype of Sox21 mutant seems not associated with SHH pathway (PMID: 19470461).

Response: We feel experimental validation is beyond the scope of this computational manuscript. The mice would have to be re-derived, which would be very time-consuming and costly. It also would require additional funding for animal and staff costs since the grant has finished. However we have taken advantage of existing HREM data to confirm the molecular red blood cell phenotype that we have identified in the *Nadk2* dataset. These data are now included in Figure 4.

Minor:

1. What does “expt” mean in fig2c?

Response: We have put the abbreviation in the figure legend.

2. Figure 2c. It is better to make clear in the legend that the samples within the circles are homozygous mutants.

Response: We have put this in the figure legend.

Reviewers' Comments:

Reviewer #1:

Remarks to the Author:

In the opinion of this reviewer the authors have substantially improved the revised manuscript. The validation of the Nadk2 mutant with regard to strong reduction or absence of erythrocytes as cause of lethality could have been more clear as the absence of blood cells in the mutant is hardly visible on the HREM photographs - staining of blood cells would help; quantitative data would be more convincing. The improved manuscript should be published.

Reviewer #2:

Remarks to the Author:

This revised work from Collins et al uses whole embryo transcriptomes of homozygous lethal mouse mutant lines at E9.5 to uncover transcriptional events underlying embryonic lethality. The revisions improve this conceptually innovative work of potential great use to the mouse genetics community. In response to the previous critique, the authors agreed that developing these descriptive data into a tool was worthwhile and report that they developed a Shiny app to allow baseline data to be used in conjunction with other E9.5 transcriptomic data. I assume that the app is Baseline CompaRe. This seems like a useful tool for comparing new data to the baseline data. The study represents an ambitious effort and potentially valuable resource. I appreciate that the authors have worked to make a more readable and rigorous study.

Reviewer #3:

Remarks to the Author:

The revised manuscript by Collins and colleagues has not changed very much. While the authors answered the reviewers' questions in the text of their rebuttal, they were not consistent about incorporating those suggestions into the revised manuscript. Accordingly, the manuscript text is still quite difficult to understand, owing to imprecise language and jargon throughout. In addition, the figures have hardly changed, which was disappointing, given the many opportunities raised in review. In its present form, I am not able to recommend this manuscript for publication, nor to endorse the quality of the and the impact of its claims.

Reviewer #4:

Remarks to the Author:

My questions have been addressed.

REVIEWERS' COMMENTS:

Reviewer #1 (Remarks to the Author):

In the opinion of this reviewer the authors have substantially improved the revised manuscript. The validation of the *Nadk2* mutant with regard to strong reduction or absence of erythrocytes as cause of lethality could have been more clear as the absence of blood cells in the mutant is hardly visible on the HREM photographs - staining of blood cells would help; quantitative data would be more convincing. The improved manuscript should be published.

Unfortunately, we do not have any quantitative data for this. To make the differences on the micrographs easier to see we have circled relevant portions of the images.

--

Reviewer #2 (Remarks to the Author):

This revised work from Collins et al uses whole embryo transcriptomes of homozygous lethal mouse mutant lines at E9.5 to uncover transcriptional events underlying embryonic lethality. The revisions improve this conceptually innovative work of potential great use to the mouse genetics community. In response to the previous critique, the authors agreed that developing these descriptive data into a tool was worthwhile and report that they developed a Shiny app to allow baseline data to be used in conjunction with other E9.5 transcriptomic data. I assume that the app is Baseline CompaRe. This seems like a useful tool for comparing new data to the baseline data. The study represents an ambitious effort and potentially valuable resource. I appreciate that the authors have worked to make a more readable and rigorous study.

Thank you.

--

Reviewer #3 (Remarks to the Author):

The revised manuscript by Collins and colleagues has not changed very much. While the authors answered the reviewers' questions in the text of their rebuttal, they were not consistent about incorporating those suggestions into the revised manuscript. Accordingly, the manuscript text is still quite difficult to understand, owing to imprecise language and jargon throughout. In addition, the figures have hardly changed, which was disappointing, given the many opportunities raised in review. In its present form, I am not able to recommend this manuscript for publication, nor to endorse the quality of the and the impact of its claims.

We have incorporated our responses to the points raised into the manuscript and have reworked the manuscript to try and improve readability.

--

Reviewer #4 (Remarks to the Author):

My questions have been addressed.

Thank you.